# Comparison of cable bacteria genera reveals details of their conduction machinery

Leonid Digel [1,2,3], Mads L Justesen [1,3,10], Nikoline S Madsen [1,3,10], Nico Fransaert[4], Koen Wouters[1,4,5], Robin Bonné[1,2], Lea E Plum-Jensen [1,2], Ian P G Marshall[1,2], Pia B Jensen [1,5], Louison Nicolas-Asselineau [1,6], Taner Drace[3,7], Andreas Bøggild[5,7], John L Hansen[5,8], Andreas Schramm [1,2], Espen D Bøjesen[5,9], Lars Peter Nielsen[1,2], Jean V Manca[4] & Thomas Boesen [1,3,5,7 ✉]

## Abstract

**Cable bacteria are centimeter-long multicellular bacteria conducting electricity through periplasmic conductive fibers (PCFs). Using single-strain enrichments of the genera *Electrothrix* and *Electronema* we systematically investigate variations and similarities in morphology and electrical properties across both genera. Electrical conductivity of different PCFs spans three orders of magnitude warranting further investigations of the plasticity of their conduction machinery. Using electron microscopy and elemental analyses, we show that the two cable bacteria genera have similar cell envelopes and cell–cell junction ultrastructures. Iron, sulfur, and nickel signals are co-localized with the PCFs, indicating key functional roles of these elements. The PCFs are organized as stranded rope-like structures composed of multiple strands. Furthermore, we report lamellae-like structures formed at the cell–cell junctions with a core layer connecting to the PCFs, and intriguing vesicle-like inner membrane invaginations below the PCFs. Finally, using bioinformatic tools, we identify a cytochrome family with predicted structural homology to known multi-heme nanowire proteins from other electroactive microorganisms and suggest that these cytochromes can play a role in the extra- or intercellular electron conduction of cable bacteria.**

**Keywords** Cable Bacteria; Electron Conduction; Electron Transfer; Electron Microscopy; Cytochrome
**Subject Categories** Microbiology, Virology & Host Pathogen Interaction; Structural Biology

## Introduction

Cable bacteria are filamentous microorganisms with an ability to conduct electrons over centimeter distances (Pfeffer et al, 2012). They are present globally in marine and freshwater sediments and are classified into two genera, *Electrothrix* and *Electronema* (Sereika et al, 2023; Dam et al, 2021; Trojan et al, 2016). Cable bacteria couple spatially separated sulfide oxidation and oxygen reduction (Pfeffer et al, 2012) via periplasmic conductive fibers (PCFs) (Cornelissen et al, 2018; Thiruvallur Eachambadi et al, 2020). The PCFs have an exceptional conductivity that can exceed 10 S/cm, which is comparable to organic semiconductors (Meysman et al, 2019; Bonné et al, 2020, 2022). The PCFs are located inside conspicuous parallel longitudinal surface ridges that span across cell junctions (Pfeffer et al, 2012). To date, little is known about the structure of the PCFs beyond their approximate dimensions and localization. The ultrastructural details of the whole conductive machinery have not been elucidated, and the elemental composition has only been reported for unidentified cable bacteria sampled from Rattekaai salt marsh (Boschker et al, 2021; Thiruvallur Eachambadi et al, 2021). Boschker and co-workers proposed a structural model, according to which cable bacteria conduct electrons through protein-based fibers with a nickel-sulfur ligated co-factor in their core and an additional protein layer forming an insulating surface of the PCFs. They found that treatment of cable bacteria with the surfactant sodium dodecyl sulfate (SDS) removed membranes and cytoplasm, but left the entire PCF network intact, bound to a polysaccharide-rich base layer. Throughout this work we term this network the "cable bacteria skeleton". Notably, previous results indicated that long-range electron transport in cable bacteria may involve nickel (Boschker et al, 2021), even though all other known biological electron transport only involves redox-active iron or copper metalloproteins (Liu et al, 2014). In other electroactive archaea and bacteria, iron-based extracellular cytochrome nanowires (ECNs) have been proposed to be the universal method of long-distance electron transport at the

[1]Center for Electromicrobiology, Aarhus University, 8000 Aarhus, Denmark. [2]Department of Biology, Aarhus University, 8000 Aarhus, Denmark. [3]Department of Molecular Biology and Genetics, Aarhus University, 8000 Aarhus, Denmark. [4]X-LAB, UHasselt, 3500 Hasselt, Belgium. [5]Interdisciplinary Nanoscience Center (iNANO), Aarhus University, 8000 Aarhus, Denmark. [6]Max Planck Institute for Marine Microbiology, 28359 Bremen, Germany. [7]EMBION - The Danish National Cryo-EM Facility – Aarhus Node, Aarhus University, 8000 Aarhus, Denmark. [8]Department of Physics and Astronomy, Aarhus University, 8000 Aarhus, Denmark. [9]Aarhus University Centre for Integrated Materials Research, Aarhus University, 8000 Aarhus, Denmark. [10]These authors contributed equally: Mads L Justesen, Nikoline S Madsen. ✉E-mail: thb@inano.au.dk

micrometer scale (Wang et al, 2019; Gu et al, 2023; Wang et al, 2022; Baquero et al, 2023). These ECNs are long heme c-based molecular wires extending from the cell surface, with electrical conductivity reaching 30 S/cm (Yalcin et al, 2020).

Conductivity studies of cable bacteria were previously done using environmental samples containing mixtures of cable bacteria strains, thus making it difficult to differentiate intra-strain and inter-strain variability, and to link results from different kinds of analyses. Recently established single-strain enrichment cultures of *Electronema aureum* GS (Thorup et al, 2021) and *Electrothrix communis* RB (Plum-Jensen et al, 2024) allowed us to perform multiple comparative analyses on the same set of cable bacteria strains. First, variations in the electrical conductivity of the filaments were measured upon direct connection to carbon paste electrodes. To investigate common and variable traits of the PCFs, we analyzed the elemental composition at nanometer-scale resolution with scanning transmission electron microscopy with energy-dispersive X-ray spectroscopy (STEM-EDX) and time-of-flight secondary ion mass spectrometry (ToF-SIMS). In addition, high-resolution cryogenic electron tomography (cryo-ET) combined with transmission electron microscopy (TEM) and scanning electron microscopy (SEM) on different cable bacteria strains and extracted PCFs were used to elucidate the ultrastructural elements of their conduction machinery. Finally, we employed comparative genomics coupled to structure prediction and homology searches to investigate metalloproteins conserved between the two cable bacteria genomes, with special focus on multiheme cytochromes and nickel-binding proteins. Combined, our findings allowed us to propose a new ultrastructural model for the cable bacteria conduction machinery.

# Results

## Cable bacteria enrichment cultures and environmental samples

For comparative studies and to ensure reproducibility we used two single-strain enrichment cultures of cable bacteria as primary source material. As a representative of the genus *Electronema*, we used a previously described enrichment of *Electronema aureum* GS (Thorup et al, 2021), here referred to as "GS". As a representative of the genus *Electrothrix* we used a recently described single-strain enrichment of *Electrothrix communis* RB, here referred to as "RB". The genome of GS was closed in 2023 (Sereika et al, 2023) and a closed genome of RB was recently published (Plum-Jensen et al, 2024). The phylogenetic position based on whole genome sequences of the two strains is shown in Fig. 1A.

For additional conductivity analyses, we also obtained cable bacteria filaments from mixed enrichment cultures derived from environmental samples, i.e., Rattekaai salt marsh (Rat (Meysman et al, 2019)) and Hou beach (Hou and Hou1) sediments. These samples contained different strains of the genus *Electrothrix* as shown by 16S rRNA gene sequencing (Fig. EV1A), from which a few representative plastic-embedded cross-sections were obtained (Fig. EV1B).

## Variability in electrical conductivity of different cable bacteria

Using carbon paste electrodes to electrically connect cable bacteria to a probe system, we measured and compared electrical conductivity of the five different cable bacteria samples: GS, RB, Rat, Hou and Hou1 (Fig. 1B). To investigate electron conduction in cable bacteria relative to other bacteria we measured the conductivity of other types of filamentous bacteria from 12 pure cultures and 3 environmental samples using interdigitated gold electrodes. Instead of connecting to carbon paste, this approach allowed us to work with fragile bacterial filaments that were pipetted onto the electrodes and dried, in contrast to cable bacteria filaments that can be picked by microscopic glass hooks. The conductivity of these non-cable bacteria species was below the detection limit of 1 pA/V, and more than 6 orders of magnitude lower than the average conductivity in cable bacteria (Appendix Table S1).

The conductivity measurements on cable bacteria were carried out under standardized conditions with a measured gap length between the carbon paste electrodes, and fixed measurement and extraction times (Fig. 1C). The measured currents were used to calculate the conductivity of single PCFs, as they are hypothesized to be the conductive structures within cable bacteria filaments. The number and the dimensions of the PCFs necessary for these calculations were also measured in this study (see below) or published previously (Cornelissen et al, 2018).

The calculated electrical conductivities, based on estimates of PCF cross-sectional area, had the same order of magnitude in GS, Hou and Rat samples (Fig. 1B) with average values of $4.0 \pm 2.8$ S/cm, $3.8 \pm 4.3$ S/cm, and $3.9 \pm 5.7$ S/cm, respectively. The highest conductivity of 25.6 S/cm was measured for a single Rat cable bacterium. In comparison, RB and Hou1 cable bacteria had 2–3 orders of magnitude lower PCF conductivity, $0.13 \pm 0.15$ S/cm, and $0.18 \pm 0.63$ S/cm, respectively. The measurements below the limit of detection were not included in the calculations. Notably, 19% of measurements on GS were below the limit of detection. In contrast, RB showed conductivity below the limit of detection in 43% of measurements. All strains tested showed linear current/voltage curves and similar stability in a nitrogen atmosphere over 72 h (for representative graphs see Appendix Fig. S1A–C). In air, conductivity decreased within 10 min (Appendix Fig. S1B) as previously observed (Meysman et al, 2019).

## Iron, sulfur, and nickel localization in periplasmic conductive fibers

Next, we set out to test whether the elemental composition of the PCFs might shed light on the differences in electrical conductivity, as well as the mechanism of electron conduction. We employed two complementary elemental analysis tools (STEM-EDX and ToF-SIMS) on GS and RB cable bacteria strains. In both strains nickel and sulfur were detected by both methods, in good agreement with the previously published data on unidentified marine cable bacteria (Boschker et al, 2021). Based on high-resolution STEM-EDX spectra, nickel signals were present in both intact cells and cable bacteria skeletons, indicating that nickel was localized in the PCFs and not in the cytoplasm, periplasm or in integral membrane proteins (Fig. 2A). Notably, the global raw signal count for iron, which is a common element in metalloproteins, was higher than the nickel signal in intact cable bacteria. After the skeleton extraction procedure, the iron signal decreased significantly relative to the nickel signal, but a clear iron signal was still present (Fig. 2A). Due to the difficulty of assigning absolute abundances of elements from STEM-EDX spectra presented in Fig. 2, only the relative

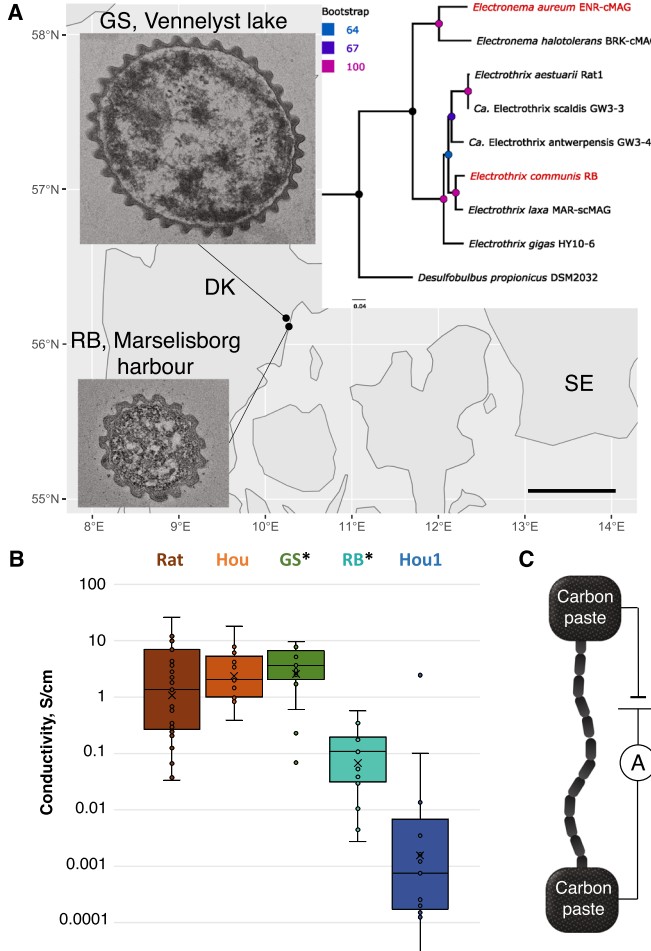

**Figure 1. Different cable bacteria strains possess variable electrical conductivity.**

(A) Original sampling sites and representative TEM images of plastic-embedded cross sections of cells of two of the investigated cable bacteria strains, *Electronema aureum* GS and *Electrothrix communis* RB. Bottom right corner shows the scale bar for the TEM images showing the difference in cell diameter and number of ridges. Scale bar: 500 nm. Panel insert: Phylogenetic tree of cable bacteria based on genomic data with the two depicted strains outlined in red. Scale bar shows the number of amino acid changes per site. (B) Electrical conductivity of PCFs of different cable bacteria calculated from measured conductance of whole bacteria and the cross-sectional average of PCFs. * mark single-strain enrichments. The bold line denotes the median (center), the lower and upper bounds of box the 25th and 75th percentiles and the whiskers extend to the minima and maxima values no larger than 1.5× inter-quartile range. Values outside of the boundary of the whiskers are outliers. Each point is a biological replicate. Rat $n = 31$, Hou $n = 17$, GS $n = 17$, RB $n = 18$, Hou1 $n = 14$. (C) Cartoon of the experimental set-up for conductance measurements of filamentous bacteria.

abundances of elements could be determined. These are depicted as relative atomic percentages for sulfur, iron and nickel in Table 1. For the PCF regions in GS cross-sections, the average relative atomic percentages of sulfur, iron, and nickel in the PCF were $24.9 \pm 10.3\%$, $72.7 \pm 11.9\%$, and $2.4 \pm 1.6\%$, respectively. For the PCF regions in RB cross-sections, these numbers were $78.6 \pm 3.3\%$ for sulfur, $10.9 \pm 1.2$ for iron, and $10.4 \pm 3.3$ for nickel. Hence, the

atomic percentage of iron in the PCF of RB is significantly lower than the iron atomic percentage observed for GS. Notably, one GS cross-section (GS cross-section #4 in Table 1) displayed relative atomic percentages of these three elements that were more similar to RB. Significant characteristic peaks were also visible from copper in the sum spectrum, especially the Kα transition at 8.04 keV. These X-rays probably originated from the sample holder and the copper grid containing the sample. Less pronounced peaks were also observed for silicon and aluminum that were attributed emissions from the silicon drift detectors within the STEM microscope.

Intact filaments of both RB and GS were also identically prepared and analyzed using the same ToF-SIMS equipment located at IMEC-Leuven (Belgium) as used in the earlier study on the unidentified marine cable bacteria (Boschker et al, 2021). The spectra of both strains clearly show the presence of signature nickel peaks (Fig. 2B), which were further confirmed by isotope analysis (Appendix Fig. S2). Moreover, the ToF-SIMS spectra of both RB and GS filaments exhibit numerous similarities (Appendix Fig. S3) and display the characteristic protein and carbohydrate signatures described previously for unidentified cable bacteria (Boschker et al, 2021).

Since the conductive core was previously proposed to be made of protein with Ni-S co-factors (Boschker et al, 2021), we looked for potential high-resolution co-localization of sulfur and nickel peaks in intact cells and extracted skeletons. STEM-EDX of intact cable bacteria detected a sulfur signal aligned along the major axis of the PCFs, as also observed for nickel and iron (Fig. 2C). To confirm that nickel and sulfur were co-localized, principal component analysis was performed in combination with non-negative matrix factorization for intact cable bacteria (Appendix Fig. S4A) and cross-sections (Appendix Fig. S4B,C). Indeed, nickel and sulfur, but also iron co-localized at the expected positions of the PCFs. Raw summed spectra of these samples can be found in Appendix Fig. S5A–C. Using the iron, sulfur, and nickel signals as an indirect measure of the PCF diameter, a value of approximately 30 nm was found from longitudinal views, which is in accordance with the PCF diameter measurements from cross sections (Table 2, Fig. 2C; Appendix Fig. S6). Finally, elemental analysis of cellular cross-sections of both GS and RB species confirmed the presence and significant signals of iron as well as nickel and sulfur co-localizing with the PCF structures (Fig. 2D,E).

## Cross-sectional TEM analysis of intact cable bacteria filaments

To improve our understanding of the ultrastructural organization of cable bacteria filaments, we collected high-resolution TEM images of cross-sections from GS and RB cable bacteria embedded in plastic (Appendix Fig. S6). The diameters of the filaments were reasonably consistent within each strain (Table 2). GS filaments were almost 2 times bigger than RB filaments in diameter (Appendix Fig. S6A,B). Consequently, GS filaments contained more than twice as many PCFs as RB: 34 vs. 15 (Table 2).

The PCFs of GS had a rounded architecture and in the regions between individual PCFs, the distance between the inner and outer membranes was, in most cases, not discernable, indicating an absence of periplasmic space in these regions (Fig. 3; Appendix Fig. S6). In contrast, the morphology of the PCFs in RB was more

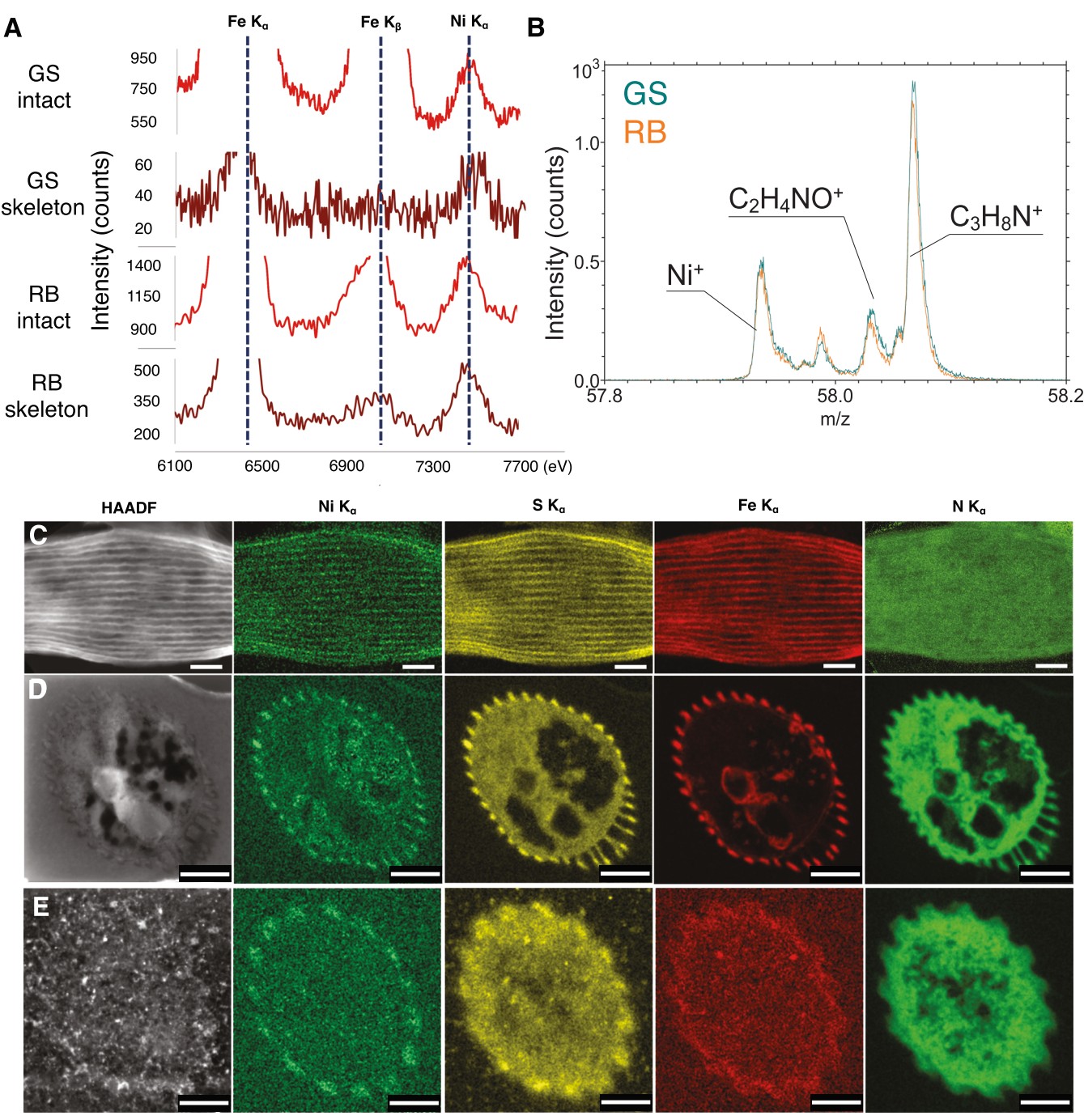

**Figure 2. Elemental composition and distribution maps of different cable bacteria.**

(A) STEM-EDX spectra of single intact filaments and skeletons of GS and RB cable bacteria. Selected characteristic X-ray energies are shown for iron and nickel. (B) ToF-SIMS spectrum showing presence of nickel in both GS and RB intact cable bacteria (represented by the cyan and orange curve, respectively). (C) High-angle annular dark-field (HAADF) image of an untreated GS cable bacterium cell and STEM-EDX images showing the distribution of nickel, sulfur, iron, and nitrogen. (D) HAADF image of one 80 nm thick cross-section of an unstained GS cable bacterium cell and STEM-EDX images showing the distribution of nickel, sulfur, iron, and nitrogen. (E) HAADF image of one 80 nm thick cross-section of an unstained RB cable bacterium cell and STEM-EDX images showing the distribution of nickel, sulfur, iron, and nitrogen. Scale bar: (C, D) 500 nm, (E) 200 nm.

rectangular, with only a few exceptions (Appendix Fig. S6). For these PCFs we measured the height and the width and from these values we calculated that the cross-sectional area of a single PCF in RB was around 50% bigger than that of GS (Table 2). Despite this

fact, the total cross-sectional area of all PCFs combined in RB was approximately 40% smaller than that of the GS (Appendix Fig. S6).

The morphology of the cell–cell junction region was of particular interest as PCFs interconnect electrically in these regions

**Table 1.** Relative amounts of sulfur, iron, and nickel in PCFs of *Electronema aureum GS* and *Electrothrix communis RB* in the STEM-EDX spectra.

| | CS # | Kα intensity (integrated counts) | | | Relative atomic percentage (%) | | | Ratio based on relative atomic percentage | | |
|---|---|---|---|---|---|---|---|---|---|---|
| | | S | Fe | Ni | S | Fe | Ni | S:Fe | S:Ni | Fe:Ni |
| GS | 1 | 651 | 5302 | 40 | 13.0 | 86.3 | 0.7 | 0.15 | 18.81 | 125.12 |
| | 2 | 373 | 944 | 52 | 31.4 | 64.9 | 3.8 | 0.48 | 8.35 | 17.25 |
| | 3 | 6559 | 17,681 | 727 | 30.3 | 66.8 | 2.9 | 0.45 | 10.50 | 23.10 |
| | 4[a] | 3956 | 1330 | 582 | 71.4 | 19.6 | 9.0 | 3.64 | 7.90 | 2.17 |
| | Average ± Std. dev. | [b] | | | 24.9 ± 10.3 | 72.7 ± 11.9 | 2.4 ± 1.6 | 0.36 ± 0.18 | 12.55 ± 5.53 | 55.16 ± 60.66 |
| RB | 1 | 1161 | 180 | 227 | 77.2 | 9.8 | 13.0 | 7.89 | 5.95 | 0.75 |
| | 2 | 1757 | 344 | 309 | 76.3 | 12.2 | 11.5 | 6.26 | 6.62 | 1.06 |
| | 3 | 36,648 | 5915 | 3493 | 82.4 | 10.9 | 6.8 | 7.59 | 12.21 | 1.61 |
| | Average ± Std. dev. | [b] | | | 78.6 ± 3.3 | 10.9 ± 1.2 | 10.4 ± 3.3 | 7.24 ± 0.87 | 8.26 ± 3.44 | 1.14 ± 0.43 |

± values represent standard deviation.

[a]Not included in average as an outlier.

[b]Average not calculated since raw counts differ based on measurement parameters/length.

(Thiruvallur Eachambadi et al, 2020) (Fig. 3A–D; Appendix Figs. S7A,B and S8). From each PCF, a lamella-like structure was observed to extend towards the center of the cell–cell junction. This structure was composed of the outer membrane and the putative surface layer on both sides of a lamella core. This lamella core displayed an internal structure showing a repetitive pattern indicating an ordered structure (Appendix Fig. S8). We termed this central structure the "core lamella sheet" (CLS) (Fig. 3B). This ultrastructural lamella component (consisting of the putative surface layer, outer membrane and CLS) is hereafter termed "junction lamella" (JL) (Fig. 3A,B,D). Intriguingly, the CLS seemed to be directly connected to the PCFs (Fig. 3A,C,D).

In the plastic-embedded cross-section images from GS and RB cable bacteria species, the cell envelope was observed to be composed of 3 layers: inner and outer membranes and a third outer layer. This third layer had not been described before and we propose that it is a surface layer (Fig. 3C,D; Appendix Fig. S6).

## Strand components of the periplasmic conductive fibers

To further investigate the structural composition of the PCFs we attempted to break the PCFs into their constituent parts. The cable bacteria skeleton is resistant to 1% SDS treatment, and previous studies confirmed that it remains electrically conductive after this treatment (Meysman et al, 2019). We found that cable bacteria filaments completely lost their integrity after harsh mechanical manipulation involving cycles of freezing and vortexing (Appendix Fig. S9A,B) or after being exposed to prolonged SDS treatment (Fig. 3E). High-angle annular dark-field (HAADF) imaging showed (Fig. 3E) that the prolonged SDS treatment had removed membranes and cytoplasm, as well as the carbohydrate layer that was suggested to keep the PCFs aligned in parallel (Boschker et al, 2021). Moreover, the PCFs themselves appeared unwound, splitting into thinner strands. We termed these "PCF strand components" and in some areas bundles of multiple strand components were observed with the smallest strands having a consistent diameter of $3.1 \pm 0.2$ nm ($n = 17$) (Fig. 3E). Similar strands or bundles were repeatedly found in samples of different cable bacteria strains using SEM and TEM (Appendix Fig. S9C).

## Cryo-ET of intact cable bacteria filaments

Cryo-ET of intact cable bacteria filaments confirmed the 3-layer envelope organization and that the PCFs contained strand components (Fig. 3F,G). Sub-tomogram averaging of the longitudinal cross-section through the surface ridge structure revealed 3D volumes of the inner membrane, a PCF with multiple strand components, an outer membrane as well as the putative surface layer (Fig. 3G). The PCF strand components did not seem to form a tight, twisted superstructure, but instead displayed a loose, parallel organization (Fig. 3G).

## Inner membrane-attached vesicles (IMAVs)

In all cable bacteria cells investigated with cryo-ET we found multiple invaginations of the inner membrane towards the cytoplasm. These vesicle-like structures (Fig. 4, Movie EV1) were predominantly observed as extensions of the inner membrane, and we therefore termed them "inner membrane-attached vesicles" (IMAVs) (Fig. 4A–E). Intriguingly, the IMAVs seemed to co-localize with the ridges containing the PCFs (Fig. 4A). Similarly sized vesicles were also observed in the cytoplasm of the cell, and these were termed Cytoplasmic Vesicles (CVs). The IMAVs, however, comprised more than 85% and up to 100% of the total vesicle count in cable bacteria cells (Appendix Table S2). IMAVs observed in GS cable bacteria varied in size with an average diameter of $65.5 \pm 4.6$ nm ($n = 90$). The CVs were similar in size to the IMAVs with an average diameter of $68.9 \pm 4.6$ nm ($n = 48$). In RB cable bacteria all vesicles were observed as IMAVs and had an average diameter of $57.6 \pm 4.8$ nm ($n = 69$) (Appendix Fig. S10).

In a few tomograms an inner membrane bulging was observed (Fig. 4C), which formed a dome-shaped structure extending into the cytoplasm from the periplasm. The contents of the domes were visibly different from the cytoplasmic content and no large structures were observed in the corresponding periplasmic matrix. Notably, IMAVs were observed connecting with the dome structures, too, and the PCFs did not seem to be affected by the bulging event, as they still appeared aligned with the ridge (Fig. 4C).

**Table 2. Dimensions of the cell structures of different cable bacteria revealed using cross-sectional electron microscopy.**

|  | Electronema aureum GS | Electrothrix communis RB |
|---|---|---|
| Filament diameter, nm | 1069 ± 94 ($n = 7$) | 613 ± 15 ($n = 6$). |
| PCF number, unitless | 34 ($n = 6$) | 15 ($n = 3$) |
| PCF dimensions, nm | D = 30.89 ± 4.05 ($n = 68$) | H = 25.95 ± 6.47 ($n = 43$) <br> W = 43.01 ± 13.5 ($n = 27$) |
| Cell–cell junction lamella, nm | 26.37 ± 1.44 ($n = 10$) | – |
| Outer membrane, nm | 5.13 ± 0.75 ($n = 10$) | – |
| Surface layer, nm | 3.82 ± 0.66 ($n = 10$) | – |
| Core lamella sheet (CLS), nm | 8.0 ± 0.6 ($n = 4$) | – |

The values are presented with ± standard deviation, where applicable.
*D* diameter, *H* height, *W* width, (–) not measured.

## Bioinformatic search for periplasmic metalloprotein candidates

Proteins involved in PCF formation were expected to be conserved among different cable bacteria genera, and we thus performed a comparative genomics analysis searching for putative iron-containing and nickel-containing proteins conserved between the GS and RB cable bacteria genomes.

16 putative nickel-binding proteins that can be grouped in four categories were identified based on sequence analysis (Dataset EV1). These were putative nickel transport proteins involved in nickel import and export, a nickel responsive regulator, nickel insertion/maturation proteins and nickel-dependent carbon monoxide dehydrogenase proteins. Of the 16 putative nickel-binding proteins only one protein (GS: CAK8712533.1) was predicted to have periplasmic localization. This protein showed homology to periplasmic transport proteins using FoldSeek and among the hits was the nickel-binding protein, NikA (Shepherd et al, 2007). Two additional nickel transport proteins were predicted to contain signal peptides but with unknown or cytoplasmic membrane localization (Dataset EV1).

We identified 11 predicted extracytoplasmic soluble c-type cytochromes that were conserved between the GS and RB genomes (Fig. 5A, Dataset EV1), in good agreement with previous genomic studies (Kjeldsen et al, 2019). Based on AlphaFold structure prediction, 10 of the cytochromes showed structural homology to previously characterized c-type cytochromes of other bacteria (Dataset EV1, Dataset EV2). The cytochrome with no structurally characterized homolog was found to be in a conserved genetic context adjacent to hypothetical proteins encoding for a trans-membrane protein, a putative iron-sulfur cluster protein and a small hypothetical protein. As no bonafide complete complex III has been identified in cable bacteria to date (Kjeldsen et al, 2019), these adjacent proteins and the tetraheme cytochrome (pACIII) could be a candidate alternative complex III containing a membrane-integrated component, iron-sulfur clusters and c-type hemes for electron transfer pathways. Based on the structural models, the c-type cytochromes were divided into two groups with the first group containing putative non-catalytic cytochromes having hexacoordinated c-type hemes. This group of cytochromes was proposed to be involved in electron transfer reactions exclusively. The second group of c-type cytochromes consisted of potential catalytic c-type cytochromes with one or more pentacoordinated hemes. The pentacoordinated hemes have an open coordination site for putative ligand binding and catalytic activity. The group of potential catalytic c-type cytochromes included MacA, pOCC, Gsu1284-like cytochrome, ExtN, sPHC, and lPHC, while the cytochromes hypothesized to exclusively transfer electrons were the Gsu1996-like cytochrome, OmcF-like cytochrome, pACIII cytochrome, cytochrome c3 and NapB (Fig. 5A, Dataset EV1). Seven of the c-type cytochromes showed domain homology to c-type cytochromes in the electroactive model organisms *Geobacter sulfurreducens* and *Shewanella oneidensis* (Dataset EV1). Interestingly, structural homology to the Geobacter ExtKL was observed for members of a specific cable bacteria pentaheme cytochrome (PHC) family (Fig. EV2A). ExtKL was proposed to be involved in extracellular electron transport in Geobacter (Salgueiro et al, 2022; Jahan et al, 2018). Furthermore, the ExtKL structural homolog of RB displayed a two-gene structure similar to that of *G. sulfurreducens*. In *G. sulfurreducens*, the *extKL* genes are located in a gene cluster that contains the *extI* gene encoding an outer membrane porin and the genes of the two multiheme cytochromes *extM* and *extN* (Jahan et al, 2018). The genomic context of the PHCs of cable bacteria differed from that of *G. sulfurreducens* and no genes encoding an outer membrane porin were found in close genomic proximity. However, we identified homologs for the ExtM and ExtN in different genomic locations. The genomes of GS and RB also encode several beta-barrel proteins that could be candidates for outer membrane porins involved in extracellular electron transfer, but these are, similarly to the ExtN and ExtM, encoded in a different genomic region than the *extKL* genes in cable bacteria.

The family of pentaheme ExtKL homologs of cable bacteria was found to form two groups based on structure prediction. One group contained a small version of the pentaheme cytochrome domain and was named small pentaheme cytochrome (sPHC). The second group contained a version of the pentaheme cytochrome with a larger cytochrome domain that may also be associated with one or two truncated hemoglobin domains and was denoted large pentaheme cytochrome with or without terminal truncated hemoglobin (lPHC/lPHCtX where X denotes that the terminal truncated hemoglobin domain may be N- or C-terminal or both) (Fig. EV2A). We observed that *G. sulfurreducens* also contains both a sPHC variant (ExtKL) and a lPHC variant, Gsu2801 (Fig. EV2B). We therefore propose that the pentaheme cytochromes of cable bacteria belong to a PHC family containing ExtKL and Gsu2801 of

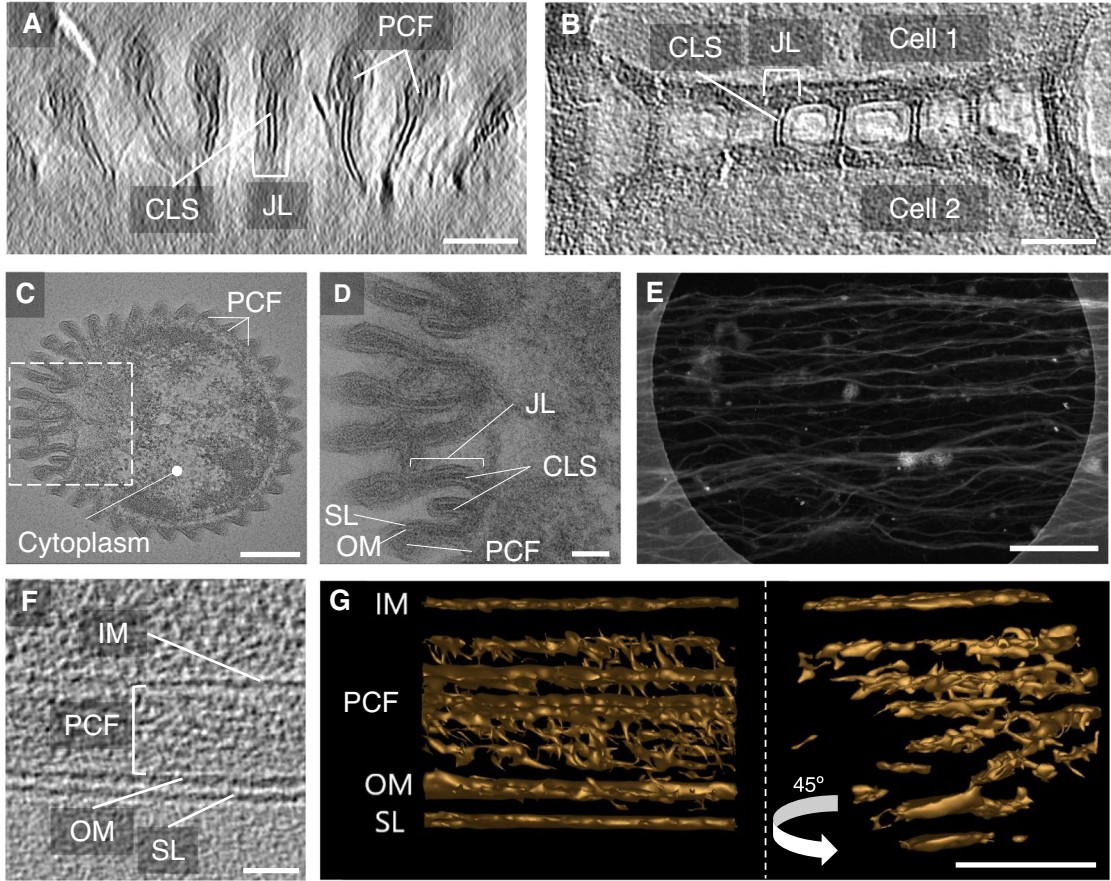

**Figure 3.  Electron microscopy investigation of the cable bacteria electron conduction machinery.**

(**A**) Cryo-ET image of a cross-sectional view of a GS cable bacterium's cell–cell junction area. (**B**) Cryo-ET image of a longitudinal section view of a GS cable bacterium's cell–cell junction lamellae. (**C**) Cross-sectional TEM image intercepting part of the cell junction of a GS cable bacterium embedded in plastic. Dashed square is the area zoomed in D. (**D**) TEM image of a cross section showing a GS cable bacterium's cell junction and envelope compartments. (**E**) HAADF image of strand components extracted from cable bacteria. (**F**) Cryo-ET image of a longitudinal section of an intact cable bacterium showing the internal structure of one PCF. (**G**) Sub-tomogram average of the PCF surroundings in F. Side (left) and near-cross-sectional (right) views of the average. CLS—core lamella sheet, JL—junction lamella structure, SL—surface layer, OM—outer membrane. Scale bar: (**A**, **B**) 100 nm, (**C**, **E**) 200 nm, (**D**) 50 nm, (**F**) 20 nm, (**G**) 30 nm.

*G. sulfurreducens*. The sPHCs of cable bacteria also showed structural homology to the *Thermochromatium tepidum* pentaheme cytochrome c552, which was recently suggested by Baquero et al to be evolutionarily related to the archaeal extracellular cytochrome nanowire AvECN (Baquero et al, 2023) (Fig. 5B). Indeed, we observed a similar heme arrangement in sPHC and AvECN with alignment of heme I, III, IV, and V of sPHC to the four hemes of AvECN (Fig. 5C). Furthermore, conservation of several secondary structural elements was observed, including similar large beta hairpin loop domains consisting of two long beta strands in the c-terminus of the RB sPHC and AvECN proteins (Fig. 5B).

## Discussion

An intriguing observation from our high-resolution ultrastructural studies was the revelation of the junction lamellae composition with a distinct core material—the CLS. Based on its localization and morphology, this is likely the material that interconnects the fibers in the cell–cell junctions making the long-distance conduction fail-

safe (Thiruvallur Eachambadi et al, 2020) (Fig. 6). Our data thus showed that the cell junction structures contain more than just two folded membranes as previously suggested (Cornelissen et al, 2018). The lack of direct evidence for the electrical conductivity in the CLS structure, and the fact that it has a structural composition distinct from the PCF makes it a very interesting target for future investigations.

Formation of cytoplasmic vesicles originating from the inner membrane in gram-negative bacteria is a poorly understood phenomenon (Toyofuku et al, 2019; Dobro et al, 2017). In *Bacillus subtilis* cytoplasmic vesicles are known to be associated with phage-induced cell death (Toyofuku et al, 2017). We observed IMAVs in cable bacteria cells that did not show signs of possible phage infection, such as phage capsids or tails (Chaikeeratisak et al, 2022). In contrast, IMAVs in cable bacteria are abundant and present in different genera, so we hypothesize that they play an important role in general cable bacteria metabolism. Küsel and co-authors speculated that inner membrane vesicles found in *Acidiphilium cryptum* were produced to increase cellular contact area with electron acceptors (Küsel et al, 1999). Photosynthetic

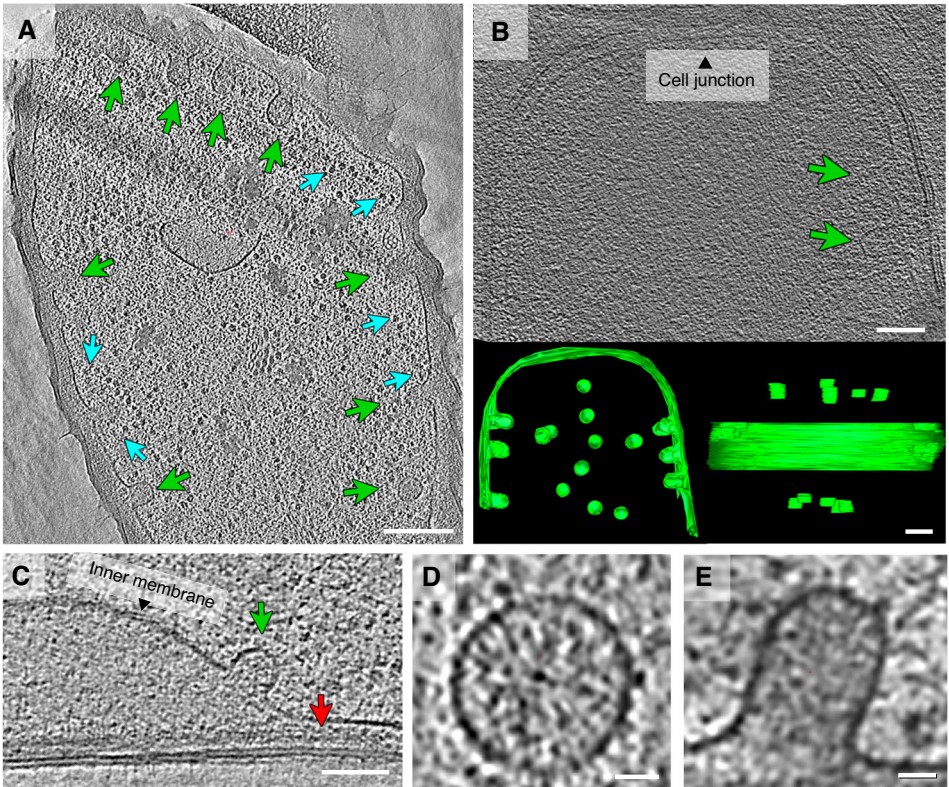

**Figure 4.  Various membrane-bound structures observed in intact cable bacteria cells.**

(A) Cryo-ET image of a near cross-section of a GS cable bacterium displaying ridge structures and multiple Inner Membrane-attached Vesicles (IMAVs). (B) Cryo-ET of a longitudinal section of an RB cable bacterium. The two figure inserts at the bottom present a top (left) and a side (right) views of a 3D segmentation of the RB inner membrane with multiple IMAVs visibly protruding into the cytoplasm. (C) Inner membrane bulging event. (D) IMAVs top view. (E) IMAV side view. Arrows: cyan—putative polysome, green—IMAV, red—PCF. Scale bar: (A) 200 nm, (B, C) 100 nm, (D, E) 20 nm.

microorganisms form chromatophores—intracytoplasmic membranes of various shapes, including vesicle-like forms—that are presumed to increase the photosynthetically-active membrane surface (Noble et al, 2018; LaSarre et al, 2018). A recent study on *G. sulfurreducens* showed that in conditions with limited energy availability they produced stacks of intracytoplasmic membranes to enhance membrane-bound processes (Howley et al, 2023). Considering that most of the vesicles that we observed in different cable bacteria were attached to the inner membrane (Appendix Table S2 and Appendix Fig. S10), we suggest that they are used as activity hotspots for membrane-dependent cellular processes involving electron transport. We thus speculate that the membrane of the IMAVs may contain electron transport chain components where exchange of electrons with the PCF is mediated by periplasmic multiheme cytochromes as suggested by Kjeldsen et al, 2019 (Fig. 6). Multiple studies have shown that components of the electron transport chain often form supercomplexes, and that these complexes can induce high curvature of the membrane as observed in the case of IMAVs (Mühleip et al, 2023; Blum et al, 2019).

The electrical current running through cable bacteria filaments is most likely conducted by the PCFs through their combined cross-sectional area. The conductivity of individual PCFs in cable bacteria was hypothesized to be highly similar, and it was suggested that thicker bacterial filaments simply form more PCFs to adjust to their higher metabolic needs. Surprisingly, our electrical conductivity

measurements found no universal conductivity value. We suggest that the quite high variability in electrical conductivity amongst the species presented here can be attributed to yet unexplored factors, such as the metabolic activity of cable bacteria at the time of sampling, active cell division or bacteriophage infections. The fact that a higher proportion of GS cable bacteria were still electrically conductive at the time of the measurement, in comparison to RB, supports this hypothesis. It was a great surprise that the periplasmic fibers appear rather amorphous without any conspicuous crystallinity based on the cryo-ET data. This raises the question of how the high conductivity observed for some cable bacteria can be obtained as opposed to the fact that high conductivity has been found to be attributed crystalline systems (Flygare and Svensson, 2021; Mott, 1969).

In earlier studies, dissection of cable bacteria filaments by an AFM probe and inspection of cross-sections with TEM (Jiang et al, 2018) revealed a dimeric structure of the PCFs in some cable bacteria, with the fiber diameter ranging between 10 and 30 nm. We show that the periplasmic fibers can be taken apart down to PCF strand components of approximately 3 nm in diameter. In addition, our data shows a consistent diameter of the PCF in GS cable bacteria of around 30 nm. Combined, this indicates that the PCFs are composed of multiple strand components forming a loose stranded rope-like structure, and that the 10 nm fibers observed previously were, likely, mechanically broken PCFs resulting in

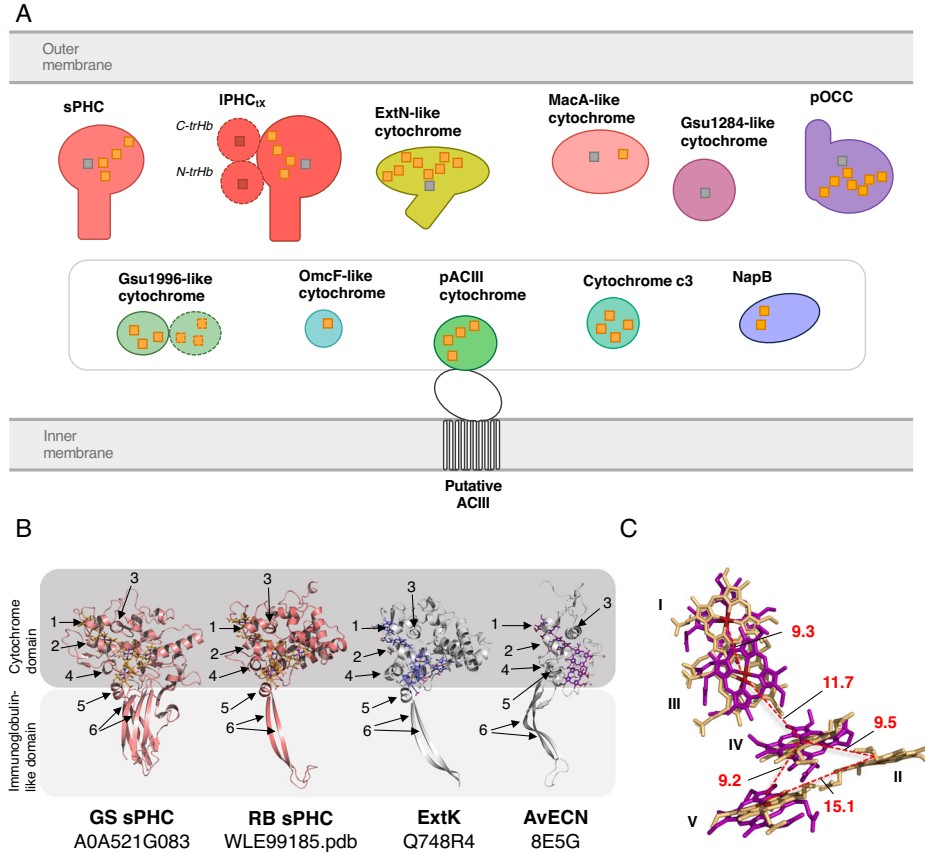

**Figure 5. Conserved periplasmic c-type cytochromes in cable bacteria and their structural homology to cytochromes involved in extracellular electron transfer.**

(A) Schematic showing the putative periplasmic c-type cytochromes, which are conserved between GS and RB. Heme c is illustrated with orange squares (hexacoordinated) and heme b with red squares. Grey squares illustrate pentacoordinated hemes. Dotted lines indicate the absence of the domain in some proteins. Proteins are named as indicated in Dataset EV1. (B) A comparison of the cable bacteria sPHCs with the ExtKL and AvECN. Arrows highlight conserved secondary structural elements numbered from N to C-terminal. Arrow nr. 6 highlights the conserved beta hairpin loop. AlphaFold models and PDB codes can be found below the structures. (C) Alignment of the RB sPHC (light orange) and AvECN (purple) heme groups. Iron-iron distances in Ångstrom are presented for the AlphaFold model of RB sPHC model. Hemes are numbered with roman numerals according to the order of heme-binding motifs in the RB sPHC sequence.

bundles of strand components. The rectangular and rounded morphologies of the surface ridges in RB and GS cable bacteria strains further support a loose oligomeric composition of the PCFs, capable of assuming various shapes.

Detection of significant amounts of iron, nickel and sulfur in the PCFs of different cable bacteria underscored their potential role either in conferring structural stability or in electron conduction. Notably, iron-sulfur metalloproteins are known to be important for electron transport in many biological systems, even though, the role of iron in electron conduction in cable bacteria has been challenged previously (Boschker et al, 2021). In the periplasmic region containing the PCFs of GS, the relative abundance of iron was up to 120 times higher than for nickel, while the relative atomic percentages of iron and nickel in the same region for RB PCFs showed a 1:1 ratio. This difference in the iron-nickel ratio and the abundance of iron might be one of the factors that can explain the observed differences in conductivity between GS and RB shown in Fig. 1B. However, in a single cross-section from GS, the relative atomic percentages of sulfur, iron, and nickel were more similar to RB. This indicates that variation in elemental composition within

the same species of cable bacteria exists, which is consistent with differences in intraspecies filament conductivity, but more work is needed to understand this putative connection. Since STEM-EDX provides no information about the chemical environment of these elements, it is uncertain whether the increased iron content of GS PCFs is due to increased levels of iron-containing proteins such as cytochromes or iron ions or minerals residing in the periplasmic ridges. Despite the low nickel-to-iron ratio in some cross-sections, nickel was reliably detected since the absolute atomic percentage of nickel in cable bacteria PCFs determined by other studies is above 0.2% (Boschker et al, 2021). This is well above the detection limit for STEM-EDX, which is around 0.05% weight percent (Kim et al, 2020). Due to the relatively high Z of nickel compared to other elements in biomolecules, the detection limit in atomic percent is expected to be even lower.

Our comparative genomic analyses did not identify a putative Ni-binding protein candidate that could be involved in PCF formation. Of the 16 nickel-binding proteins, 12 proteins have recently been identified in the context of nickel homeostasis (Hiralal et al, 2024). The NikA homolog was the only candidate

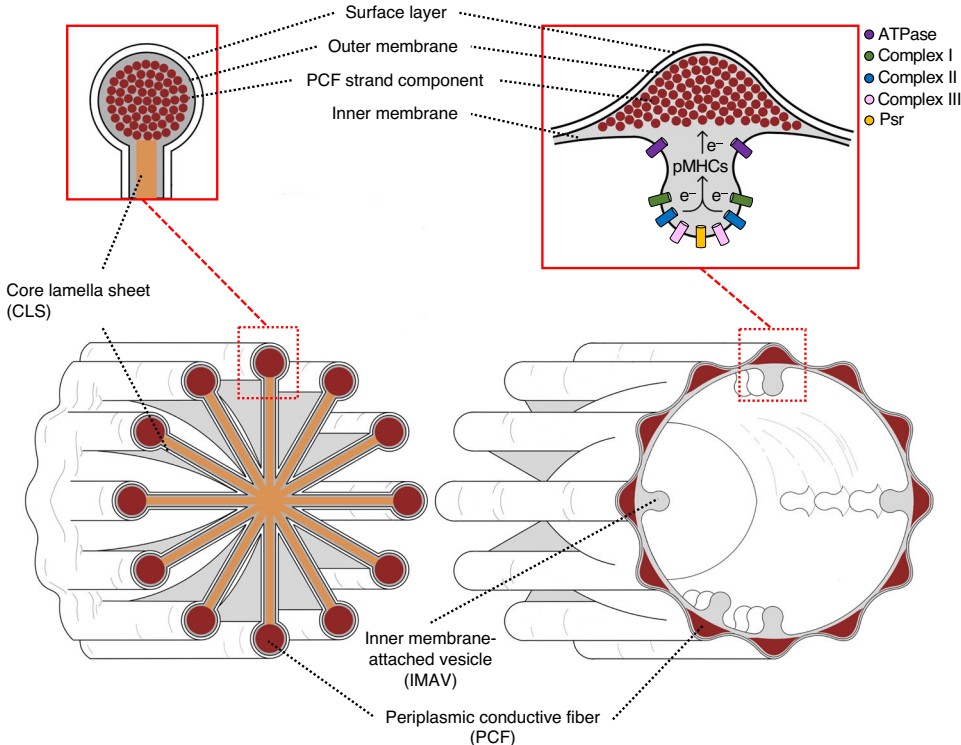

**Figure 6.   A new morphological model of the cable bacteria cell.**

The model is presented as a cross-sectional view of the cell junction (left) and the middle of the cell (right). Top left corner insert represents the PCF organization at the cell junction, and the connection to the putatively conductive CLS. Top right corner insert represents the PCF organization in the middle of the cell, and the suggested association with the IMAVs via periplasmic multiheme cytochromes (pMHCs). Organization of hypothesized electron transport chain complexes was reconstructed based on data from (Kjeldsen et al, 2019). ATPase adenosine triphosphate synthase, Psr polysulfide reductase.

nickel-binding protein, identified in this study, that has a predicted periplasmic location and could be contributing to the observed elemental nickel signal in the PCFs. In *E. coli*, it was found to bind heme as well as nickel suggesting it could play a role incorporating heme into cytochromes (Shepherd et al, 2007). There is no available data suggesting that NikA can form filaments and have a role in electron conduction. This, however, does not exclude the possibility of another nickel-containing protein giving rise to the signal in the periplasm. With the homology-based approach employed here we would not be able to identify an uncharacterized protein/motif belonging to a new family of nickel-binding proteins.

In our search for periplasmic iron-binding proteins, we identified 11 putative periplasmic c-type cytochrome proteins conserved between the two cable bacteria genomes used in this study, which represent two different genera (Dataset EV1). Six of these cytochromes—pACIII cytochrome, MacA, pOCC, Gsu1284-like cytochrome, lPHCtX and OmcF-like cytochrome—were found to be expressed in previous proteomic studies even with incomplete proteome coverage (Kjeldsen et al, 2019). Since the study of Kjeldsen and colleagues, there have been significant advancements in protein structure prediction with the release of artificial intelligence-driven tools like AlphaFold. The bioinformatic approach used in this study provided a better foundation for characterization of the cable bacteria cytochromes using the more conserved structural homology rather than sequence homology, with a specific focus on electroactive bacteria (Illergård et al, 2009).

Intriguingly, among the structurally conserved cytochromes, a family of pentaheme cytochrome proteins was suggested to have structural homology to nanowire-forming multi-heme cytochromes in Archaea based on AlphaFold predictions and database searches. To our knowledge this is the first time that a putative nanowire-forming c-type cytochrome has been shown to be encoded in cable bacteria genomes and, in combination with our observation that iron is consistently found to be associated with the PCFs, this finding puts forward the possibility that PCFs could have a multi-heme cytochrome component responsible for long-distance electron conduction and/or performing a catalytic function (Kjeldsen et al, 2019; Digel et al, 2023). Structural homology of sPHC was observed to the low potential C552 cytochrome from the thermophilic purple sulfur photosynthetic bacterium *T. tepidum*. No clear functional role has been assigned to this cytochrome, but it has been suggested to play a role in sulfur metabolism (Chen et al, 2019). Other members of the cable bacteria PHC family were already described by Kjeldsen and coworkers in 2019 (Kjeldsen et al, 2019). The lPHCtX proteins were described as proteins where a multiheme cytochrome domain was fused to one or two truncated hemoglobin domains and this was shown to be a unique protein domain composition only found in cable bacteria. As no canonical oxidase was identified in the genomes analyzed by Kjeldsen and coworkers, it was suggested that this fusion protein (lPHCtX) could accept electrons from the PCFs and acts as an alternative terminal oxidase in the periplasm based on previous studies showing

bioelectrocatalytic oxygen reduction activity of truncated hemoglobins (Kjeldsen et al, 2019; Fernandez et al, 2013). We recently demonstrated that cable bacteria skeletons have a reversible oxygen reduction/oxidation activity which indicates a link between the suggested nanowire structure and oxygen reduction capacity of PHC family members and the PCFs (Digel et al, 2023).

Cable bacteria are the only microorganisms known to conduct electrons over centimeter distances. Their metabolism is truly unique and even having their complete genomes has not allowed us to reliably reconstruct their metabolism, since the function of many genes in cable bacteria is unknown and important canonical metabolic genes were not identified previously (Kjeldsen et al, 2019; Sereika et al, 2023). This study uncovered multiple novel cellular structures that appear unique to cable bacteria and conserved across both cable bacteria genera, and allowed us to put forward a new ultrastructural model of the conductive machinery of cable bacteria involving IMAVs and CLS as novel elements (Fig. 6). The exact molecular nature of the PCFs remains unclear, and there appears to be a difference in relative amounts of iron and nickel. The ultrastructural and compositional variability of the conduction machinery in cable bacteria emphasizes once again the complexity of the underlying electrical transport mechanisms. Although the direct connection between the physico-chemical architecture and the observed difference in the electrical conductivity is not yet clear, this study provides valuable elements for further interdisciplinary investigations. The discovery that iron co-localizes with sulfur and nickel in PCFs, as well as identification of putative nanowire-forming pentaheme cytochromes opens for a composite structure of the PCFs containing both nickel- and iron-binding proteins and highlights the importance for future high-resolution structural characterization of the hitherto unknown electrically conductive biomaterial of cable bacteria.

# Methods

### Reagents and tools table

| Reagent/Resource | Reference or Source | Identifier or Catalog Number |
|---|---|---|
| **Experimental models** | | |
| *Electronema aureum GS* | Center for Electromicrobiology | GenBank assembly GCA_942492785.1 |
| *Electrothrix communis RB* | Center for Electromicrobiology | GenBank assembly GCA_030644725.1 |
| **Recombinant DNA** | | |
| None | | |
| **Antibodies** | | |
| None | | |
| **Oligonucleotides and other sequence-based reagents** | | |
| Primer Bac341F (CCTACGGGNGGCWGCAG) | Scholz et al, 2021 and Herlemann et al, 2011 | – |
| Bac805R (GACTACHVGGGTATCTAATCC) | Scholz et al, 2021 and Herlemann et al, 2011 | – |

| Reagent/Resource | Reference or Source | Identifier or Catalog Number |
|---|---|---|
| **Chemicals, Enzymes and other reagents** | | |
| ddH$_2$O | Merck Millipore | Q-POD® Ultrapure Water Remote Dispenser |
| Sodium dodecyl sulfate (SDS) | Sigma-Aldrich | Product no.: 74255-250G |
| Ethylenediaminetetraacetic acid (EDTA), purified grade, 99% | Sigma-Aldrich | Product no.: ED-1KG |
| Water-soluble carbon paste EM-Tec C33 | Micro to Nano | Product no.: RS-MN-15-001133 |
| Sodium cacodylate buffer 0.2 M, pH 7.4 | AMPLIQON | Catalogue no.: AMPQ40989.0500 |
| 2.5% formaldehyde/2.5% glutaraldehyde in 0.1 M sodium cacodylate buffer, pH 7.4. | Electron Microscopy Sciences | Catalogue no.: 15949 |
| Ethanol absolute | VWR Chemicals | Catalog no.: 20821.310 |
| Propylene oxide | Sigma-Aldrich | Product no.: 110205-100 ML |
| EPON-815 RESIN | Electron Microscopy Sciences | Product no.: 14910 |
| Uranyl Acetate Solution | Electron Microscopy Sciences | Product no.: 22400-1 |
| Agarose | Merck | Product no.: 101236 |
| Lead(II) nitrate | Merck | Catalogue no.: 107398 |
| Collodion solution | Sigma-Aldrich | Product no.: 1.02644 |
| **Software** | | |
| IMOD 4.12.21 | Kremer et al, 1996 | – |
| IsoNet | Liu et al, 2022 | – |
| MotionCor2 | Zheng et al, 2017 | – |
| PEET 1.16.0 | Nicastro et al, 2006; Heumann et al, 2011 | – |
| InterPro 101.0 | Paysan-Lafosse et al, 2023 | – |
| PSORTb v3.0 | Yu et al, 2010 | – |
| SignalP 5.0 | Almagro Armenteros et al, 2019 | – |
| FoldSeek | van Kempen et al, 2023 | – |
| PyMOL Version 2.5.4 | The PyMOL Molecular Graphics System, Version 2.5.4 Schrödinger, LLC | – |
| DALI protein structure comparisons server | Holm et al, 2023 | – |
| AlphaFold 2.3.1 | Jumper et al, 2021 | – |

| Reagent/Resource | Reference or Source | Identifier or Catalog Number |
|---|---|---|
| GTDB-Tk tool (v. 2.2.3) | Chaumeil et al, 2020 | – |
| GTDB-Tk IQ-tree (v. 1.6.12) | Nguyen et al, 2015 | – |
| **Other** | | |
| Electron microscopy grids Au-Flat. GF-2/2-2Au-45nm-50. | Protochips | Product no.: SKU: M-CEM-AUFT222-50 |
| Electron microscopy grids C-Flat. CF-2/2-2Cu-50. | Protochips | Product no.: SKU: CF-222C-50 |
| Electron microscopy grids C-Flat. CF-2/2-3Cu-50. | Protochips | Product no.: SKU: CF-223C-50 |
| Colloid gold, 10 nm | BBI Solutions | Product no.: EM.GC10 |
| Trench slides | Plum-Jensen et al, 2024 | – |
| Cover slips | Hounisen Laboratorieudstyr A/S | Product no.: 0422.2460 |
| DNeasy PowerSoil Pro Kit | Sereika et al, 2023 | – |
| Qubit dsDNA HS kit | Sereika et al, 2023 | – |
| Circulomics SRE XS kit | Sereika et al, 2023 | – |
| DNeasy PowerLyzer PowerSoil Kit | Scholz et al, 2021 | – |

## Cable bacteria strains

Single-strain enrichment cultures of *Electronema aureum* GS (GCA_942492785.1) (Thorup et al, 2021) and *Electrothrix communis* RB (GCA_030644725.1) (Plum-Jensen et al, 2024), established from local freshwater and coastal sediments, respectively, and maintained in our laboratory were used for investigating defined cable bacteria species. In addition, mixed cable bacteria enrichments were established and maintained in our laboratory from coastal sediment collected at Hou beach, Hou, Denmark (55.915086, 10.256861) for samples Hou and Hou1, and for Rattekaai cable bacteria (Rat) from Rattekaai salt marsh sediment, collected in Rilland, The Netherlands (51.439167, 4.169722).

## DNA extraction and sequencing

DNA extraction and metagenome sequencing of the Hou sediment samples has been described previously (Sereika et al, 2023) (Reagents and Tools Table); DNA extraction and 16S rRNA gene amplicon sequencing of the Rattekaai sediment samples was done according to the protocol of (Scholz et al, 2021) (Reagents and Tools Table).

## Bioinformatic methods for identification of periplasmic metalloprotein candidates

The genomes of *Electronema aureum GS* and *Electrothrix communis RB* were searched for the presence of proteins with nickel and iron cofactors that could explain the co-localization of these elements with the periplasmic conductive fibers. Candidate

proteins were selected if they were 1. conserved between species of cable bacteria, 2. predicted to be periplasmic or extracytoplasmic.

Nickel binding proteins present in the genome were identified using sequence level annotation and potentially nickel binding domains identified using InterPro 101.0 (Paysan-Lafosse et al, 2023). Cellular localization was predicted using PSORTb v3.0 (Yu et al, 2010) and SignalP 5.0 (Almagro Armenteros et al, 2019). The presence of a protein in both species of cable bacteria was based on sequence homology using Basic Local Alignment Search Tool for protein-protein (Protein BLAST) (Altschul et al, 1990). The CAK8712533.1 AlphaFold database structure (AF-A0A521G2F4-F1) was run in FoldSeek (van Kempen et al, 2023) against the PDB100 database. Homology to characterized NikA structures was evaluated in PyMOL Version 2.5.4 (The PyMOL Molecular Graphics System, Version 2.5.4 Schrödinger, LLC.).

To identify c-type cytochrome candidates that could be responsible for the observed iron signals the genomes were searched for proteins with CXXCH motifs associated with heme c binding. Cellular localization of the c-type cytochromes was predicted using PSORTb v3.0 (Yu et al, 2010) and SignalP 5.0 (Almagro Armenteros et al, 2019). AlphaFold structure predictions of the identified c-type cytochromes of GS were found in the AlphaFold Protein Structure Database through UniProt accession (Jumper et al, 2021; Varadi et al, 2022; The Uniprot Consortium, 2023). AlphaFold structure predictions for c-type cytochromes identified in RB were generated using Parafold for AlphaFold 2.3.1 (Jumper et al, 2021; Zhong et al, 2022; Arnold, 2021). Structurally characterized homologs were identified using the DALI protein structure comparisons server (Holm et al, 2023) and FoldSeek (van Kempen et al, 2023) on AlphaFold structure models. The structural models of the identified cable bacteria c-type cytochromes were compared and aligned in PyMOL Version 2.5.4 (The PyMOL Molecular Graphics System, Version 2.5.4 Schrödinger, LLC.). Structural alignments to Geobacter cytochromes without characterized structures were performed in PyMOL Version 2.5.4 to AlphaFold structure models from the AlphaFold Protein Structure Database. Note the new CAK8714958.1 and the old TAA75631 accession numbers for ExtM.

## Phylogenetic analysis

Published genomes of cable bacteria were used for phylogenetic analysis. The 120 conserved bacterial single copy genes (Parks et al, 2018) were identified with the GTDB-Tk tool (v. 2.2.3, (Chaumeil et al, 2020)). The single copy genes were aligned with the GTDB-Tk and a phylogenetic tree was made with IQ-tree (v. 1.6.12, (Nguyen et al, 2015)) with 1000 Bootstraps and *Desulfobulbus propionicus DSM2032* as outgroup.

## Bacterial filament extraction

Clean cable bacteria filaments were obtained by placing a piece of the sediment under a stereomicroscope and fishing cable bacteria out of the sediment by a home-made glass hook. The filaments were washed in drops of sterile filtered Milli-Q water to remove sand and dust particles.

To extract the cable bacteria skeletons, the cable bacteria were washed as described above and then were moved to another droplet with 1% weight/volume SDS for 10 min, washed with Milli-Q water

up to three times, treated with 1 mM EDTA, pH 8, for 10 min, washed again and air dried. These extracted and purified PCFs were lastly moved to the appropriate substrate for further investigations, for example, Electron Microscopy grids, silicon substrate for EDX, or carbon paste electrodes.

To produce PCF strand components from the PCFs, alternative treatments were used. Clean cable bacteria filaments were placed on EM grids and 4 μL of 1% SDS were added on the grid to completely cover its surface (Reagents and Tools Table). After 100 min of incubation at room temperature, the samples were washed with 100–150 μL of MilliQ water by blotting from the bottom of the grid. The grid was directly taken for analysis. Mechanical extraction of strand component structures included 3 cycles of freezing in liquid nitrogen and vortexing, and centrifugation at $12,000 \times g$ for 10 min.

## Conductivity measurements

Conductivity of cable bacteria was measured using a 4200 Keithley-SCS (Keithley, Solon (Ohio), USA). Sample sizes were as follows: Rat—31, Hou—17, GS—17, RB—18, Hou1—14. Cable bacteria filaments were fished out of the sediment and washed in MilliQ water at least 6 times before being placed on a glass substrate. The filament was dried with nitrogen gas for 30 s, before carbon paste (EM-Tec C33, Micro to Nano, Haarlem, the Netherlands) was applied on both ends of the filament. All of this was conducted within a 5-min timeframe. The gap size (= the distance between the carbon paste electrodes) was between 100 and 800 μm as measured afterwards with a light microscope. The prepared sample was placed in a probe stage, which was flushed with $N_2$ twice before being placed in (low) vacuum. Then, Tungsten/Iridium probes were mounted on the carbon paste electrodes and voltage scans of $-1$ to 1 V were conducted, while the current was monitored. From the linear I/V curves, the conductivity was calculated using $\sigma = (I \cdot l)/(V \cdot A)$ with $I$ the current at $V = 100 mV$, $l$ the length of the conductive channel, and $A$ the cross-sectional area of the PCFs. The cross-sectional area was calculated from the diameter of the fibers, assuming one fiber to be 850 nm² for Rat, Hou, Hou1, and GS, and 1100 nm² for RB. Since Hou and Hou1 were not single-strain enrichments, the number of PCFs was estimated from the diameter of the filaments measured with a light microscope. Assuming a linear correlation between the filament diameter and the number of PCFs, the following formula was used: PCF number = $5 \times \pi \times$ filament diameter (Source data file for Fig. 1B; Appendix Fig. S1). The number of PCFs in Rat cable bacteria was published previously as an average value measured from different filaments (Cornelissen et al, 2018).

All control samples were prepared by placing one to hundreds of bacterial filaments on an interdigitated Au electrode with 50 μm interdistance, and dried (PW4XIDEAU50, Metrohm, Oviedo, Spain). The sample was placed in a vacuum atmosphere to prevent possible degradation and ensure adhesion of the filament to the electrode surface. The same system as described above was used to estimate the conductivity of 15 selected filamentous bacteria (Appendix Table S1). Cultures were grown as recommended by DSMZ (Deutsche Sammlung von Mikroorganismen und Zellkulturen) or described in the original publication listed in the Appendix Table S1. Now I is the cumulative current from all 10

parallel current paths, l the conductive channel (50 μm), and A the cross-sectional area of the bacterial filament (0.5 μm²). This last one is a conservative lower limit assuming 5 filaments with a similar cross-sectional area to those of cable bacteria. With currents never exceeding the pA range, conductivities were always found to be below $10^{-7}$ S/cm. In most cases, more filaments were applied, ending up in conductivities below $10^{-8}$ S/cm. This was repeated thrice for every species ($n = 3$), with one negative control being a dried drop of the culture medium, which gave currents in the same pA range.

## STEM-EDX

STEM-EDX spectra of intact cable bacteria, PCF-skeletons, and cross-sections were recorded on an FEI Talos FX200i field emission gun transmission electron microscope (ThermoFisher Scientific) operated in scanning transmission mode. The sample was placed on a gold electron microscopy grid with 200, 2/2 mesh. Individual spectra and element distribution maps were acquired within 20 and 560 min at 33,000–94,000 times magnification with a beam current of 0.472 to 7.48 nA and a beam energy of 200 keV and a dwell time of 20.0 μs. Velox (ThermoFisher Scientific) was used to collect the raw data and produce binned elemental maps and Hyperspy (GitHub project https://doi.org/10.5281/zenodo.7263263) was used to extract raw summed spectra and for principal component analysis and non-negative matrix factorization. For determination of relative atomic percentages in PCFs, a polygonal selection was made on the spectrum image to select an area corresponding to the PCF, and a sum spectrum was extracted for each of these areas. Nine and ten areas/PCFs were selected and averaged for GS and RB, respectively.

## ToF-SIMS

Multiple cleaned intact filaments of cable bacteria were deposited on a gold-coated silicon wafer. The filaments were then located by both fluorescence microscopy and AFM for subsequent ToF-SIMS analysis. ToF-SIMS data was acquired using the TOF.SIMS 5 device (IONTOF GmBH, Germany) located at Imec, Leuven (Belgium). The high current bunched mode was used for maximal mass resolution with a $Bi_3^+$ analysis beam (30 keV, current ~0.35 pA, $100 \times 100$ μm² area, $512 \times 512$ pixels). Existing factory settings were optimized to obtain maximal current. This setup is traditionally employed for surface spectrometry; however, despite being slower than a dedicated sputter gun, the $Bi_3^+$ analysis beam could still penetrate through the filaments, and this arrangement ensured the collection of all filament material. Data analysis was performed using the SurfaceLab 7 software (IONTOF GmBH, Germany). All measurements were calibrated using $C^+$, $C_2H_3^+$, $C_3H_4^+$, $C_3H_5^+$, $C_4H_5^+$ and $Au^+$. Mass spectra were obtained by summing over all pixels in the lateral region of interest (ROI) and over all data points in the depth ROI: substrate regions were excluded from the lateral ROI to minimize substrate counts, and the first ten data points were removed from the depth ROI since these data points are related to surface transient. The relative count of isotope $^xNi$ was calculated as the ratio of its absolute count to the sum of the absolute counts of the three most abundant nickel isotopes $^{58}Ni$, $^{60}Ni$, and $^{62}Ni$. The expected counts were calculated

based on the natural abundances of $^{58}$Ni (68.08%), $^{60}$Ni (26.22%), and $^{62}$Ni (3.64%) (Appendix Fig. 2S).

## Cross-sectional TEM

Cable bacteria filaments were fished from the sediment and washed in a water droplet to remove dust and sand particles. Clean bacteria were quickly injected into a small drop of agarose and dumped into the fixative 0.1 M sodium cacodylate buffer solution containing 2.5% formaldehyde and 2.5% glutaraldehyde and stored in the refrigerator for three days. Afterwards, the sample was rinsed twice with 0.1 M sodium cacodylate buffer for 10 min, rinsed 3 times with dH$_2$O for 5 min, and rinsed 2 times for 10 min with increasing concentrations of ethanol as follows; 50%, 70%, 90%, and 99%; the 99% ethanol rinse was done 2 times for 15 min. The last washing solution was propylene oxide, 2 times for 15 min. The sample was then moved to a mixture of propylene oxide 1:1 in EPON epoxy resin and left overnight under the fume hood. The agarose drops with the cable bacteria filaments were transferred from the propylene oxide/Epon mixture into pure Epon for 8 h, then placed in a mold and left to cure at 60 °C for at least 48 h. The resulting Epon block was then put into the microtome for sectioning by cutting with a glass knife for an overview, while thin sections were cut with a diamond knife at an angle of 45°. Selected 50–60 nm sections were put on homemade copper EM grids with 300 mesh size and covered with 2% collodion solution and a film of carbon using carbon evaporator. Post stain á La Reynolds (Reagents and Tools Table) was used to prepare the sections for TEM data collection with TEM Tecnai Spirit (ThermoFisher Scientific).

## Sample preparation for FIB-SEM and cryo-ET

To extract clean cable bacteria filaments from our sediment enrichment cultures we used diffusion gradient chambers called trench slides (Plum-Jensen et al, 2024). They consisted of a microcopy slide with a cavity in the middle, into which 3 to 5 mm$^3$ of sediment volume were inoculated. Then the slide was covered with a cover slip, which limits oxygen supply into the sediment and forces cable bacteria to move towards the edge of the slide in search of oxygen. In these chambers they were incubated for 24 h at room temperature and high humidity to limit evaporation. After 24 h the cover slip was removed, and clean cable bacteria filaments were pipetted onto electron microscopy grids just before plunge freezing (Reagents and Tools Table). 3 µl of gold beads were added as fiducials into the selected GS samples. All cable bacteria samples for cryogenic investigation were plunge frozen on 2/2 200 gold mesh grids with a gold film using Leica EM GP2 or GP1 plunge freezer (Leica Microsystems, Germany) at 10 °C and 90% relative humidity. 63% Propane/37% ethane mix was used as a cryogen. The samples were stored in liquid nitrogen.

## FIB-SEM

After rapid fixation by plunge freezing, cable bacteria that were thicker than 600 nm in diameter (GS and Hou) were milled with a Cryo-Focused-Ion-Beam Scanning Electron Microscope (Cryo-FIB-SEM) Aquilos 2 (ThermoFisher Scientific) to a thickness of approximately 250 nm, which allowed for collection of cryo-ET data of a cellular volume slice with good contrast. Two layers of

platinum were deposited on the grid to protect the sample and ensure conductivity. Milling was performed manually (without automated milling) with a gallium ion beam. Multiple tilt-series for cryo-ET were collected from the resulting "windows" into the cells (see the next section). This approach allowed us to see the morphological details in intact – non-stained – cable bacteria filaments in a frozen hydrated state.

## Cryo-ET

Cable bacteria samples were prepared as described above and milled with FIB-SEM or in their intact form in the case of RB. Tilt series were collected on a Titan Krios G3i (Thermo Fischer Scientific, USA), 300 kV, equipped with a Gatan BioQuantum/K3 energy filter and camera setup. The dose was set to 2 e$^-$/Å$^2$ for data collection on lamellae or 2.3 e$^-$/Å$^2$ for data collection on intact cable bacteria at magnification of 26,000×. The estimated pixel size was 3.385 Å with the defocus values ranging from 3.00 to 10.00 µm. Tilt series of cable bacteria lamellae were collected dose-symmetrically from −60 to 60 degrees with increments of 2 degrees. Movies were motion-corrected using MotionCor2 (Zheng et al, 2017). Tomograms were then reconstructed with eTomo as a part of IMOD 4.12.21 (Kremer et al, 1996) and CTF-deconvoluted using IsoNet (Liu et al, 2022). Inner membrane and IMAV segmentation was done manually using the modeling toolkit of 3dmod version 4.11.25. PEET 1.16.0 was used for sub-tomogram averaging on raw tomograms (Nicastro et al, 2006; Heumann et al, 2011). The sub-tomogram average of the PCF and ridge compartment (Fig. 3G) was produced from 34 sub-tomograms, resulting in a resolution of 2.4 nm (FSC = 0.5).

## Data availability

All data needed to evaluate the conclusions are present in the main text or the Appendix. The Source data files are available at BioStudies repository https://doi.org/10.6019/S-BSST1751, via the following link: https://www.ebi.ac.uk/biostudies/studies/S-BSST1751?key=054061a0-f17d-4261-b4af-7a2115870ae8.

The source data of this paper are collected in the following database record: biostudies:S-SCDT-10_1038-S44319-025-00387-8.

## Peer review information

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

## Acknowledgements

We want to thank Ronny Baaske, for establishing the single-strain enrichment of Electrothrix communis RB, Jesper L. Wulff, Susanne Nielsen, Britta Poulsen, Marie B. Lund, and Mette L. Nikolajsen for excellent technical support. We thank Federico Aulenta of Italian National Research Council and Caterina Levantesi who provided us with three pure cultures of filamentous bacteria for conductivity measurements. We thank the Danish cryo-EM Facility (EMBION) for access to electron microscopes and laboratories (funded by the Danish Ministry for Higher Education and Science grant no. 5072-00025B and Novo Nordisk Foundation grant no. NNF20OC0060483), and Jesper Lykkegaard Karlsen for assistance with bioinformatic tools. We thank Xuya Yu and Mingdong Dong who assisted with probe stage conductivity measurements, and Lars Damgaard and Jesper J. Bjerg for discussions on the subject. We also thank Benjamin Nash from the University of East Anglia for AlphaFold Heme script V5. Danish National Research Foundation (Danmarks Grundforskningsfond) grant DNRF136 (LPN), Research Foundation – Flanders (Fonds Wetenschappelijk Onderzoek) FWO project grant G013922N (JM). Research Foundation – Flanders FWO PhD Fellowship grant 11K4322N (NF).

## Author contributions

**Leonid Digel**: Conceptualization; Data curation; Formal analysis; Validation; Investigation; Visualization; Methodology; Writing—original draft; Writing—review and editing. **Mads L Justesen**: Conceptualization; Formal analysis; Investigation; Visualization; Methodology; Writing—review and editing. **Nikoline S Madsen**: Formal analysis; Investigation; Visualization; Methodology; Writing—review and editing. **Nico Fransaert**: Conceptualization; Formal analysis; Investigation; Visualization; Methodology; Writing—review and editing. **Koen Wouters**: Conceptualization; Investigation; Methodology; Writing—review and editing. **Robin Bonné**: Formal analysis; Validation; Investigation; Visualization; Methodology; Writing—review and editing. **Lea E Plum-Jensen**: Investigation; Visualization; Writing—review and editing. **Ian P G Marshall**: Supervision; Writing—review and editing. **Pia B Jensen**: Investigation; Visualization; Writing—review and editing. **Louison Nicolas-Asselineau**: Investigation; Writing—review and editing. **Taner Drace**: Investigation; Writing—review and editing. **Andreas Bøggild**: Investigation; Writing—review and editing. **John L Hansen**: Supervision; Writing—review and editing. **Andreas Schramm**: Supervision; Methodology; Writing—review and editing. **Espen D Bøjesen**: Conceptualization; Supervision; Methodology; Writing—review and editing. **Lars Peter Nielsen**: Conceptualization; Supervision; Funding acquisition; Writing—review and editing. **Jean V Manca**: Conceptualization; Supervision; Funding acquisition; Methodology; Writing—review and editing. **Thomas Boesen**: Conceptualization; Investigation; Supervision; Funding acquisition; Methodology; Writing—review and editing.

Source data underlying figure panels in this paper may have individual authorship assigned. Where available, figure panel/source data authorship is listed in the following database record: biostudies:S-SCDT-10_1038-S44319-025-00387-8.

## Disclosure and competing interests statement

The authors declare no competing interests.

# Expanded View Figures

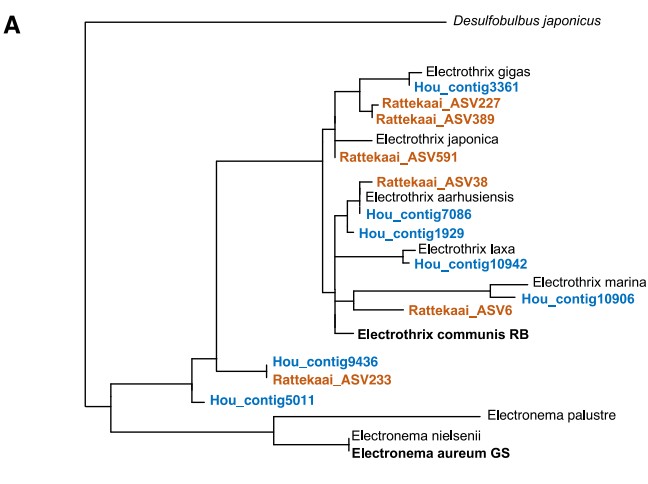

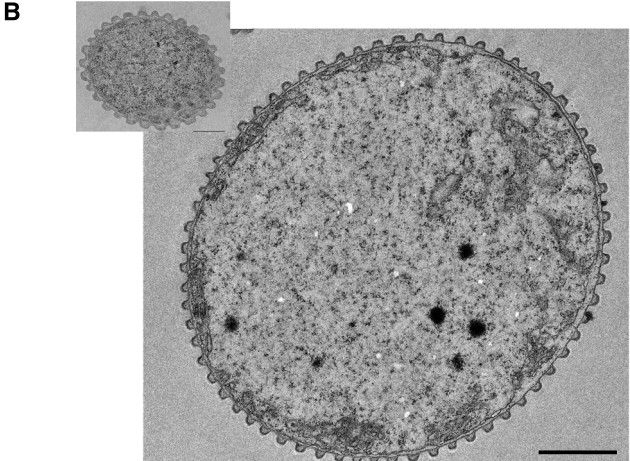

**Figure EV1. Cable bacteria diversity and morphology in the additional samples used for conductivity measurements and based on 16S rRNA gene sequencing.**

(A) Blue, sequences extracted from metagenome sequencing of Hou beach. Orange, data from amplicon sequencing of Rattekaai sediment. Black, reference sequences, with single-strain cable bacteria cultures in bold. A maximum likelihood tree was calculated in ARB using near-complete 16S rRNA gene sequences, and partial (ASV) sequences were added using the maximum likelihood tool without changing the tree topology. The phylogenetic analysis showed that both marine sites exclusively contained cable bacteria of the genus Electrothrix and indicates at least 5 different Electrothrix species in the Rattekaai samples, and at least 7 different Electrothrix species in the Hou beach samples. Scale bar, 0.1 base changes per site. (B) Representative TEM images of plastic-embedded cross sections of cable bacteria cells sampled from Hou beach (top) and Rattekaai sediment (bottom). Scale bar: 1 μm.

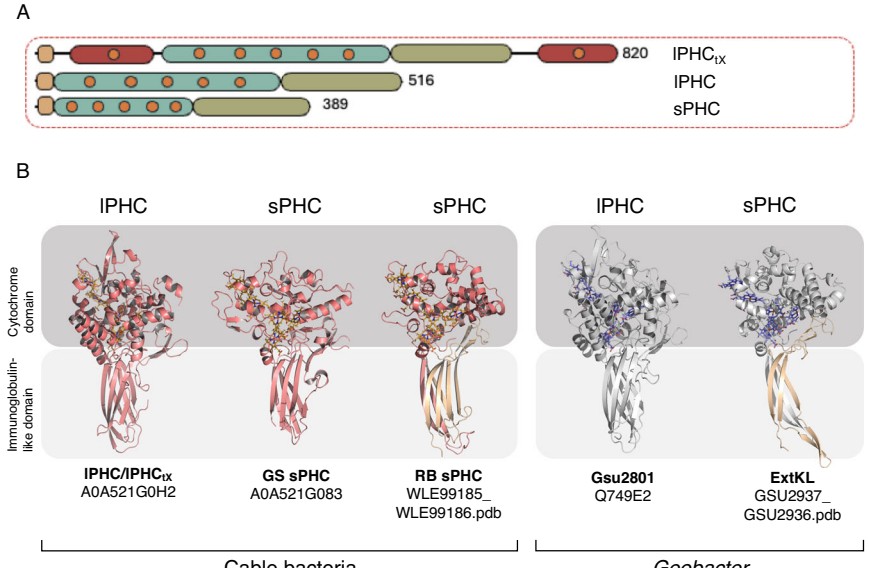

**Figure EV2. The conserved pentaheme cytochrome (PHC) family.**

(A) Schematic of the domain structure of PHC family members from GS cable bacteria. Orange circles—hemes, red—truncated hemoglobin domain, green—immunoglobulin-like domain, cyan—PHC domain, yellow square—signal peptide. (B) The structural models of the proposed pentaheme cytochrome family, that is conserved in both cable bacteria and *Geobacter*. Folding of the downstream genes (wheat) for RB sPHC and ExtKL results in a predicted structure with the larger immunoglobulin-like domain and similar overall fold to the other PHCs.

