## [Peer Review File · EMBO Reports]

Comparison of cable bacteria genera reveals details of their conduction machinery

Leonid Digel, Mads Justesen, Nikoline Madsen, Nico Fransaert, Koen Wouters, Robin Bonn , Lea Plum-Jensen, Ian Marshall, Pia Jensen, Louison Nicolas-Asselineau, Taner Drace, Andreas B ggild, John Hansen, Andreas Schramm, Espen B jesen, Lars Nielsen, Jean Manca, and Thomas Boesen

Corresponding author(s): Thomas Boesen (thb@inano.au.dk)

Review Timeline:

Submission Date:	11th Jul 24
Editorial Decision:	13th Aug 24
Revision Received:	13th Dec 24
Editorial Decision:	16th Jan 25
Revision Received:	29th Jan 25
Accepted:	30th Jan 25

Editor: Achim Breiling

Transaction Report:

Dear Dr. Boesen,

Thank you for the submission of your manuscript to EMBO reports. I have now received the reports from the three referees that were asked to evaluate your study, which can be found at the end of this email.

As you will see, the referees find the study interesting. The referees have several comments, concerns, and suggestions, indicating that need to be addressed to allow publication of the study in EMBO reports. As the reports are below, and all the concerns need to be addressed, I will not detail them further here. It seems referee #3 has seen a previous version of the manuscript at a journal outside of EMBO press. He indicates that the version now submitted to us has improved and her/his previous concerns have been adequately addressed. The referee has some minor remaining points that I also ask you to address during revision.

Acceptance of your manuscript will depend on a positive outcome of a second round of review. It is EMBO reports policy to allow a single round of revision only and acceptance of the manuscript will therefore depend on the completeness of your responses included in the next, final version of the manuscript.

- 1) a .docx formatted version of the final manuscript text (including legends for main figures, EV figures and tables), but without the figures included. Figure legends should be compiled at the end of the manuscript text.
- 2) individual production quality figure files as .eps, .tif, .jpg (one file per figure), of main figures and EV figures. Please upload these as separate, individual files upon re-submission.

4) a complete author checklist, which you can download from our author guidelines

(<https://www.embopress.org/page/journal/14693178/authorguide>). Please insert page numbers in the checklist to indicate where the requested information can be found in the manuscript. The completed author checklist will also be part of the RPF.

5) that primary datasets produced in this study (e.g. RNA-seq, CHIP-seq, structural and array data) are deposited in an appropriate public database. If no primary datasets have been deposited, please also state this in a dedicated section (e.g. 'No primary datasets have been generated and deposited'), see below.

The accession numbers and database should be listed in a formal "Data Availability" section (placed after Materials & Methods) that follows the model below. This is now mandatory (like the COI statement). Please note that the Data Availability Section is restricted to new primary data that are part of this study. This section is mandatory. As indicated above, if no primary datasets have been deposited, please state this in this section

Data availability

8) Regarding data quantification and statistics, please make sure that the number "n" for how many independent experiments were performed, their nature (biological versus technical replicates), the bars and error bars (e.g. SEM, SD) and the test used to calculate p-values is indicated in the respective figure legends (also for EV figures and all those in an Appendix). Please also check that all the p-values are explained in the legend, and that these fit to those shown in the figure. Please provide statistical testing where applicable. Please avoid the phrase 'independent experiment', but clearly state if these were biological or technical replicates. Please also indicate (e.g. with n.s.) if testing was performed, but the differences are not significant. In case n=2, please show the data as separate datapoints without error bars and statistics. See also: <http://www.embopress.org/page/journal/14693178/authorguide#statisticalanalysis>

9) Please add scale bars of similar style and thickness to microscopic images, using clearly visible black or white bars (depending on the background). Please place these in the lower right corner of the images themselves. Please do not write on or near the bars in the image but define the size in the respective figure legend.

10) Please also note our reference format:

12) We now use CRediT to specify the contributions of each author in the journal submission system. CRediT replaces the author contribution section. Please use the free text box to provide more detailed descriptions and do NOT provide your final manuscript text file with an author contributions section. See also our guide to authors: <https://www.embopress.org/page/journal/14693178/authorguide#authorshipguidelines>

13) All Materials and Methods need to be described in the main text using our 'Structured Methods' format, which is required for all research articles. According to this format, the Materials and Methods section should include a Reagents and Tools Table (listing key reagents, experimental models, software, and relevant equipment and including their sources and relevant identifiers), uploaded as separate file, followed by a Methods and Protocols section in which we encourage the authors to describe their methods using a step-by-step protocol format with bullet points, to facilitate the adoption of the methodologies across labs. More information on how to adhere to this format as well as downloadable templates (.doc) for the Reagents and Tools Table can be found in our author guidelines (section 'Structured Methods'):

14) Please order the manuscript sections like this, using these names:

Title page - Abstract - Keywords - Introduction - Results - Discussion - Methods - Data availability section - Acknowledgements including funding information) - Disclosure and Competing Interests Statement - References - Figure legends - Expanded View Figure legends

I look forward to seeing a revised form of your manuscript when it is ready.

Yours sincerely,

Referee #1:

The work described by Digel and co-authors provides a very interesting and scientifically sound observations on cable bacteria. These bacteria are electrically conductive and conduit electricity through periplasmic conductive fibres (PCFs) at centimetre scales. The fibres present very high conductivity and given their length open important avenues for eventual practical applications. The study focuses mainly on single-strain enrichment of the genera *Electrothrix* and *Electronema* that allowed the authors to surpass an important bottleneck associate with the used of environmental samples typically containing mix-cultures of cable bacteria, which have impaired through the years to differentiate between intra- and inter-strain variability amongst other key features. The quality of the work and the new findings here reported warrants its publication and seed important routes for future interdisciplinary investigations in the field.

By combining several complementary techniques, the authors show that the electrically conductivity is very different in cable bacteria. This apparently cannot be correlated with the newly identified junction lamellae, which also revealed distinct composition amongst the targeted cable bacteria. Another relevant discover was the observation of the formation of cytoplasmic vesicles originating from the inner membrane. On this topic, I suggest to the authors to develop the possible organization of the putative electron transfer chain components embedded in these structures and how they interplay with the other morphological structures discussed in this manuscript.

Finally, the discovery that iron co-localizes with sulfur and nickel in PCFs, suggest an important role for iron-binding proteins in the conductive properties of cable bacteria. On the other hand, the genomic analysis carried out by the authors seems to exclude the presence of Ni-binding protein in the formation of the PCF. This is a point of debate within the community. Instead, the presence of c-type multiheme cytochromes were revealed from this study. The authors refer to a pentaheme cytochrome with homology to the *Geobacter* ExtKL and ExtN cytochromes. Given the predicted cellular localization of ExtKL and ExtN cytochromes, the authors should discuss this in more detail, within the context of cable bacteria, the putative role of the pentaheme cytochrome including the rational for the association of this cytochrome with truncated hemoglobin domains in the same species.

In the particular case of the small version of the pentaheme cytochrome (sPHC) its putative role, compared to the structural

homolog pentaheme cytochrome c552, should also be addressed in more detail. In Figure 5, an additional panel containing only the heme groups and respective distances (iron-iron or edge-edge) will be very useful. This is not clear enough from what is presented in panel C.

Referee #2:

This study performs some of the first high resolution imaging using single-isolate cable bacteria cultures, in many cases confirming past observations but under conditions where the data can be conclusively linked to a single organism/genome. A recent observation finding nickel associated with cables is repeated, although quantification of this signal remains elusive. TEM and especially Cryo-ET imaging reveals new features I have not seen elsewhere, such as structures at the cell:cell interface, vesicles, and evidence that 'cables' are made up of smaller ~3 nm parallel fibers. Finally, some genome gazing produces some speculation about cytochromes, perhaps the most notable being their similarity to subunits of Archaeal nanowires. No data is provided or referenced for whether these cytochromes are known to be expressed based on prior data, so these discussions remain at a basic level.

I appreciate the skepticism throughout the discussion and lack of definitive conclusions that often plagues the conductivity field. This work presents new but understandably qualitative information, in a form that others can go out and verify for themselves.

1. P3. It is welcome to see some effort to provide 'controls' for long-bacterium conductivity. But, it is not clear the conductivity of 'non-cable bacteria' can be compared to measurements with cable bacteria. Cable bacteria were fixed to a probe system with carbon paste and a slight vacuum. There are zero additional methods describing the IDAs, but text suggests 'filamentous' bacteria were pipetted onto IDAs and maybe dried (?) and no carbon paste or other approaches were used (?). Provide measurements of cable bacteria tested via the same method or warn readers *in the main text* of differences with the paste/dried/probe systems.

2. P3. Data from 'Rat' and 'Hou', (apparently enrichments of some unknown age), are in the first figure, but all information about them is buried in supplementary. No images are provided to indicate how cross section is calculated, or how similar they are to the RB/GS samples. While they are lab enrichments, data in Fig S1 says it is metagenome sequencing of environmental sediments. Provide accurate descriptions of Rat and Hou (as representative cross sections and on the phylogenetic tree in the main figure/text of the paper).

3. P3. "The measured currents were used to calculate the conductivity of single PCFs, as they are the conductive structures within cable bacteria filaments". This has to read "...as they are hypothesized to be the conductive structures..." -- while it is clearly a reasonable hypothesis, we have been down this road before where something was pointed to as *the conductive thing* and it turned out to be totally wrong.

4. P3. By the same token, correct language in other sentences to reflect this current is being projected into these filaments *mathematically* by some estimate: "The electrical conductivities of the PCFs had the same..." becomes "The calculated electrical conductivities, based on estimates of PCF cross sectional area, had the same...", can correct P7 "The electrical current running through cable bacteria filaments" to "... is **most likely** conducted by..."

P3. Appreciate the honesty in showing instances where samples were below the limit of detection.

5. P3. Elemental/STEM-EDX analysis. It is helpful that this tries to compare and verify with recent studies, and reports similar findings. But, now that a signal can be found, how is a reader supposed to know what a 'strong' signal is? S is a common component of protein, so bands of N and S suggest protein, but everything seems adjusted for the max intensity, so for all one knows there is 100x more S in these images than Ni. There is nothing to compare to. Provide some kind of quantitative information.

The high noise level in the Ni images suggests these are cranked up in sensitivity; without more information it can mean the sensitivity keep changing until we get the image we want. Are we talking sensitivity of 0.1 at%? 10 ppm? What would lead us to think this is a *lot* of Ni or S? Can the ratio of S:N, S:Ni or S:Fe counts be expressed, at least based on the settings used, to show there is an enrichment?

Fig S6. There sure seems to be a lot of Cu and Fe in these samples, based on raw EDS counts which is supposed to correlate with quantity, but this is not discussed. The x-axes are different for B and C vs. A, making it harder to align common peaks.

6. P5. No information is given to support statements like "No putative nickel-sulfur periplasmic fiber candidate proteins were identified from the GS and RB genomes." - what were the criteria? Signal peptides? Tat signals? Binding motifs? Searching for the word "Nickel-Sulfur"? List the Pfam or Interpro motifs or HMM patterns searched for.

7. P6. This statement is false: "The periplasmic c-type cytochrome network of cable bacteria thereby appears to resemble that of

Geobacter sulfurreducens but contains fewer members." There are no homologs of any Geobacter quinone oxidoreductases which are genetically and biochemically shown to be responsible for feeding electrons into the periplasm. However, the ACIII cytochrome looks more like it is part of a novel NrfCD-family quinone oxidoreductase one could speculate is a pmf-dependent link between the quinone and periplasmic pools. These haven't been described in Geobacter.

There are no homologs of the most abundant triheme periplasmic cytochromes also shown to be functional in vivo in Geobacter. The PpcA (triheme)/1996 (dodecaheme) homolog looks to be a hexaheme of similar fold, but there is no direct homolog in Geobacter.

There are no homologs of periplasmic cytochromes proven to be part of functional porin-cytochrome complexes. What is present, beyond a bunch of monoheme (common in many organisms) and diheme (common peroxidases), and possibly catalytic/nitrogen cycling cytochromes (common octaheme family) are some cytochromes that are most interesting in their similarity to the Archaeal nanowire subunits. The ExtIKLMN system is biochemically shown to be an outer membrane porin-cytochrome complex in Geobacter, linked not to anything periplasmic but instead to electron transfer across the outer membrane. The presence of multiple ExtIKL-like cytochromes, with their little hydrophobic beta-sheet dangling out, yet no apparent porin/outer membrane component, is very un-Geobacter, highlighting how different the periplasmic network is, rather than similar.

8. It would be immensely helpful to readers who lack access to all the old locus tags and data to check if homologs of any of these proteins are highly abundant or highly expressed in any past transcriptional/proteomic experiments. It would strengthen the case for getting interested in any of these if there was evidence they were 1) present in all or most cable genomes, and 2) were known to be expressed. If it's not a nickel highway and they are nanowires running somewhere in the cell, a lot of protein would be needed.

Referee #3:

I have reviewed this article in a different journal. This draft has improved since the last submission, and my previous concerns have been addressed. I only have one final point:

Under STEM-EDX, Ni, S, and Fe have nearly identical signals. While it's intriguing that the author identified potential cytochrome nanowire homologs, the absence of Ni-S protein candidates raises questions. Were additional elements measured to confirm the abundance of only Ni and S? Additionally, are there any sequence motifs for Ni-S proteins that could be searched for within the genome?

Point-by-point reply to the reviewers EMBOR-2024-59924V1

We thank the Reviewers for constructive critique. We have generally revised the manuscript and below you can find our point-by-point replies to the comments.

Referee #1:

The work described by Digel and co-authors provides a very interesting and scientifically sound observations on cable bacteria. These bacteria are electrically conductive and conduit electricity through periplasmic conductive fibres (PCFs) at centimetre scales. The fibres present very high conductivity and given their length open important avenues for eventual practical applications. The study focuses mainly on single-strain enrichment of the genera *Electrothrix* and *Electronema* that allowed the authors to surpass an important bottleneck associate with the used of environmental samples typically containing mix-cultures of cable bacteria, which have impaired through the years to differentiate between intra- and inter-strain variability amongst other key features. The quality of the work and the new findings here reported warrants its publication and seed important routes for future interdisciplinary investigations in the field.

By combining several complementary techniques, the authors show that the electrically conductivity is very different in cable bacteria. This apparently cannot be correlated with the newly identified junction lamellae, which also revealed distinct composition amongst the targeted cable bacteria. Another relevant discover was the observation of the formation of cytoplasmic vesicles originating from the inner membrane. On this topic, I suggest to the authors to develop the possible organization of the putative electron transfer chain components embedded in these structures and how they interplay with the other morphological structures discussed in this manuscript.

Reply: We thank the reviewer for this suggestion! We have now integrated the previously identified putative electron transport chain components from the publication of (Kjeldsen et al, 2019) into the Figure 6, and schematically represented how electrons could be transferred from the electron transport chain, towards the pool of periplasmic cytochromes (pMHCs) and the periplasmic conductive fibers (PCFs). The legend of the Figure 6 was rewritten accordingly.

Finally, the discovery that iron co-localizes with sulfur and nickel in PCFs, suggest an important role for iron-binding proteins in the conductive properties of cable bacteria.

On the other hand, the genomic analysis carried out by the authors seems to exclude the presence of Ni-binding protein in the formation of the PCF. This is a point of debate within the community. Instead, the presence of c-type multiheme cytochromes were revealed from this study. The authors refer to a pentaheme cytochrome with homology to the *Geobacter* ExtKL and ExtN cytochromes. Given the predicted cellular localization of ExtKL and ExtN cytochromes, the authors should discuss this in more detail, within the context of cable bacteria,

Reply: the Discussion and the Results sections have now been rewritten to expand on the potential roles of ExtKL and ExtN cytochromes. Please see the lines 289-300, and 419-427.

the putative role of the pentaheme cytochrome including the rationale for the association of this cytochrome with truncated hemoglobin domains in the same species. In the particular case of the small version of the pentaheme cytochrome (sPHC) its putative role, compared to the structural homolog pentaheme cytochrome c552, should also be addressed in more detail.

Reply: We thank the reviewer for this suggestion! Lines 427-441 now contain a more extensive discussion of the putative roles of these cytochromes.

In Figure 5, an additional panel containing only the heme groups and respective distances (iron-iron or edge-edge) will be very useful. This is not clear enough from what is presented in panel C.

Reply: The Figure 5 and its legend were modified accordingly. The new panel C now contains only the heme groups and iron-iron distances.

Referee #2:

This study performs some of the first high resolution imaging using single-isolate cable bacteria cultures, in many cases confirming past observations but under conditions where the data can be conclusively linked to a single organism/genome. A recent observation finding nickel associated with cables is repeated, although quantification of this signal remains elusive. TEM and especially Cryo-ET imaging reveals new features I have not seen elsewhere, such as structures at the cell:cell

interface, vesicles, and evidence that 'cables' are made up of smaller ~3 nm parallel fibers. Finally, some genome gazing produces some speculation about cytochromes, perhaps the most notable being their similarity to subunits of Archaeal nanowires. No data is provided or referenced for whether these cytochromes are known to be expressed based on prior data, so these discussions remain at a basic level.

Reply: Lines 410-415 now contain additional text and a reference to the previously published expression data. Dataset S1, too, was modified and now contains an extra column, in which cytochromes are linked to the publication of Kjeldsen et al 2019.

I appreciate the skepticism throughout the discussion and lack of definitive conclusions that often plagues the conductivity field. This work presents new but understandably qualitative information, in a form that others can go out and verify for themselves.

1. P3. It is welcome to see some effort to provide 'controls' for long-bacterium conductivity. But, it is not clear the conductivity of 'non-cable bacteria' can be compared to measurements with cable bacteria. Cable bacteria were fixed to a probe system with carbon paste and a slight vacuum. There are zero additional methods describing the IDAs, but text suggests 'filamentous' bacteria were pipetted onto IDAs and maybe dried (?) and no carbon paste or other approaches were used (?). Provide measurements of cable bacteria tested via the same method or warn readers *in the main text* of differences with the paste/dried/probe systems.

Reply: we thank the reviewer for their comment. The required information was now moved from the Supplementary Information document to the Methods section of the main text, lines 557-570. Please see the lines 110-113, which explain the rationale for the use of IDAs to inform the readers.

2. P3. Data from 'Rat' and 'Hou', (apparently enrichments of some unknown age), are in the first figure, but all information about them is buried in supplementary. No images are provided to indicate how cross section is calculated, or how similar they are to the RB/GS samples. While they are lab enrichments, data in Fig S1 says it is metagenome sequencing of environmental sediments. Provide accurate descriptions of Rat and Hou (as representative cross sections and on the phylogenetic tree in the main figure/text of the paper).

Reply: Rat and Hou samples were indeed enrichments of cable bacteria, but were not single-strain enrichments and contained a mix of different *Electrothrix* cable bacteria. We were thus reluctant to insert cross-sections into the main text, because it is hard to justify that they are representative of an enrichment culture containing multiple species.

We have now added two cross-sections from Rat and Hou samples to the new Figure EV1, which will be displayed in the main HTML of the paper in a collapsible format, as examples of what kinds of filaments can be found in those samples. However, they can only be used as rough estimates. The data that we now deposited as Source Data for Figure 1B (numerical data – please see Data Availability section) contains a table that lists size measurements of Hou cable bacteria, which were used to estimate the number of PCFs for individual measurements. The size and the number of PCFs in Rat were reported previously and now the text contains proper citations. Lines 550-556 in the Methods section now contain more details on our calculations.

3. P3. "The measured currents were used to calculate the conductivity of single PCFs, as they are the conductive structures within cable bacteria filaments". This has to read "...as they are hypothesized to be the conductive structures..." -- while it is clearly a reasonable hypothesis, we have been down this road before where something was pointed to as *the conductive thing* and it turned out to be totally wrong.

Reply: the text in the respective places was rewritten accordingly

4. P3. By the same token, correct language in other sentences to reflect this current is being projected into these filaments *mathematically* by some estimate: "The electrical conductivities of the PCFs had the same...' becomes "The calculated electrical conductivities, based on estimates of PCF cross sectional area, had the same...", can correct P7 "The electrical current running through cable bacteria filaments" to "... is ****most likely**** conducted by..."

Reply: the text in the respective places was rewritten accordingly

P3. Appreciate the honesty in showing instances where samples were below the limit of detection.

5. P3. Elemental/STEM-EDX analysis. It is helpful that this tries to compare and verify with recent studies, and reports similar findings. But, now that a signal can be found, how is a reader supposed to know what a 'strong' signal is? S is a common component of protein, so bands of N and S suggest protein, but everything seems adjusted for the max intensity, so for all one knows there is 100x more S in these images than Ni. There is nothing to compare to. Provide some kind of quantitative information.

The high noise level in the Ni images suggests these are cranked up in sensitivity; without more information it can mean the sensitivity keep changing until we get the image we want. Are we talking sensitivity of 0.1 at%? 10 ppm? What would lead us to think this is a *lot* of Ni or S? Can the ratio of S:N, S:Ni or S:Fe counts be expressed, at least based on the settings used, to show there is an enrichment?

Reply: We thank the reviewer for this comment. To allow for comparison of relative amounts of Ni, S and Fe signals we have put together a new Table 2, which presents relative atomic percentages of these elements. We have to note, however, that STEM-EDX data in Figure 2 present elemental distribution maps and cannot be used for absolute quantification, but we added extra text and references to give the readers a sense of how sensitive the techniques is. Lines 393-398.

Fig S6. There sure seems to be a lot of Cu and Fe in these samples, based on raw EDS counts which is supposed to correlate with quantity, but this is not discussed. The x-axes are different for B and C vs. A, making it harder to align common peaks.

Reply: The x-axis was adjusted based on the reviewer's suggestion (new Figure S5, previously S6). Cu is a present in our STEM-EDX instrument and is not originating from the sample. This point is now written in the main text lines 154-158.

6. P5. No information is given to support statements like "No putative nickel-sulfur periplasmic fiber candidate proteins were identified from the GS and RB genomes." - what were the criteria? Signal peptides? Tat signals? Binding motifs? Searching for the word "Nickel-Sulfur"? List the Pfam or Interpro motifs or HMM patterns searched for.

Reply: Thank you for your comment! To address this point, we put together a new table to Dataset S1, which contains a list of putative nickel-binding proteins. We also added more text to elaborate on how this list was made. Lines 258-267, 399-409, and 483-488

7. P6. This statement is false: "The periplasmic c-type cytochrome network of cable bacteria thereby appears to resemble that of *Geobacter sulfurreducens* but contains fewer members." There are no homologs of any *Geobacter* quinone oxidoreductases which are genetically and biochemically shown to be responsible for feeding electrons into the periplasm. However, the ACIII cytochrome looks more like it is part of a novel NrfCD-family quinone oxidoreductase one could speculate is a pmf-dependent link between the quinone and periplasmic pools. These haven't been described in *Geobacter*.

There are no homologs of the most abundant triheme periplasmic cytochromes also shown to be functional *in vivo* in *Geobacter*. The PpcA (triheme)/1996 (dodecaheme) homolog looks to be a hexaheme of similar fold, but there is no direct homolog in *Geobacter*.

There are no homologs of periplasmic cytochromes proven to be part of functional porin-cytochrome complexes. What is present, beyond a bunch of monoheme (common in many organisms) and diheme (common peroxidases), and possibly catalytic/nitrogen cycling cytochromes (common octaheme family) are some cytochromes that are most interesting in their similarity to the Archaeal nanowire subunits. The ExtIKLMN system is biochemically shown to be an outer membrane porin-cytochrome complex in *Geobacter*, linked not to anything periplasmic but instead to electron transfer across the outer membrane. The presence of multiple ExtKL-like cytochromes, with their little hydrophobic beta-sheet dangling out, yet no apparent porin/outer membrane component, is very un-*Geobacter*, highlighting how different the periplasmic network is, rather than similar.

Reply: We thank the reviewer for this well-founded comment. The section has now been rewritten accordingly and the statement "The periplasmic c-type cytochrome network of cable bacteria thereby appears to resemble that of *Geobacter sulfurreducens* but contains fewer members." was removed to avoid confusion.

8. It would be immensely helpful to readers who lack access to all the old locus tags and data to check if homologs of any of these proteins are highly abundant or highly

expressed in any past transcriptional/proteomic experiments. It would strengthen the case for getting interested in any of these if there was evidence they were 1) present in all or most cable genomes, and 2) were known to be expressed. If it's not a nickel highway and they are nanowires running somewhere in the cell, a lot of protein would be needed.

Reply: Thank you for pointing this out. The Dataset S1 now contains extra columns with the information/references related to the past proteomic/transcriptomic experiments.

Referee #3:

I have reviewed this article in a different journal. This draft has improved since the last submission, and my previous concerns have been addressed. I only have one final point:

Under STEM-EDX, Ni, S, and Fe have nearly identical signals. While it's intriguing that the author identified potential cytochrome nanowire homologs, the absence of Ni-S protein candidates raises questions. Were additional elements measured to confirm the abundance of only Ni and S?

Reply: We thank the reviewer for their comment. To resolve the problem of Ni, S and Fe signals appearing identical in STEM-EDX we have put together a new Table 2, which presents relative atomic percentages of these elements. We have to note, however, that STEM-EDX data in Figure 2 present elemental distribution maps and cannot be used for absolute quantification. Extra text was added to the manuscript to strengthen this point.

Additionally, are there any sequence motifs for Ni-S proteins that could be searched for within the genome?

Reply: Thank you for your comment! To address this point, similar to the question from Reviewer 2, we put together a new table to Dataset S1, which contains a list of putative nickel-binding proteins. We also added more text to elaborate on how this list was made. Lines 258-267, 399-409, and 483-488.

Dear Dr. Boesen,

Thank you for the submission of your revised manuscript to our editorial offices. I have now received the report from one of the two referees that I asked to re-evaluate the study, you will find below. As you will see, the referee #1 now fully supports the publication of the study in EMBO reports. Referee #2 was completely unresponsive to my invitations to re-assess the study. However, going through your p-b-p-response, I consider the points of this referee (and also the final point by referee #3) as adequately addressed.

Before I can proceed with formal acceptance, I have these editorial requests I ask you to address in a final revised manuscript:

- Please provide a comprehensive final title with not more than 100 characters (including spaces).
- Please provide the abstract with not more than 175 words and written in present tense throughout.
- Please order the manuscript sections like this, using (only) these names:
Title page - Abstract - Keywords - Introduction - Results - Discussion - Methods - Data availability section - Acknowledgements - Disclosure and Competing Interests Statement - References - Figure legends - Expanded View Figure legends
- Please make sure that all figure panels (main, EV and Appendix figures) are called out separately and sequentially. Presently, there are no callouts for panels 4D and 4E. Please check.
- Please check again that the number "n" for how many independent experiments were performed, their nature (biological versus technical replicates), the bars and error bars (e.g. SEM, SD) and the test used to calculate p-values is indicated in the respective figure legends (main and EV figures). Please also check that all the p-values are explained in the legend, and that these fit to those shown in the figure. Please provide statistical testing where applicable. Please avoid the phrase 'independent experiment', but clearly state if these were biological or technical replicates. Please also indicate (e.g. with n.s.) if testing was performed, but the differences are not significant. In case n=2, please show the data as separate datapoints without error bars and statistics. See also:
<http://www.embopress.org/page/journal/14693178/authorguide#statisticalanalysis>

If $n < 5$, please show single datapoints for diagrams. Moreover:

- Please note that the box plots need to be defined in terms of minima, maxima, centre, bounds of box and whiskers, and percentile in the legend of figure 1B.
- Please note that information related to n is missing in the legend of figure 1B.
- Please add a proper table of contents (TOC) to the first page of the Appendix file mentioning each item with its corresponding page number. Please remove the legends and mentions of movies and datasets from the Appendix. The Appendix file should only contain the Appendix items and their legends.
- Please name the two datasets 'Dataset EV1' and 'Dataset EV2'. Please update file names, titles, legends and manuscript callouts accordingly. Please provide their legends as a separate tab/sheet in the Excel file, or as a README file in the ZIP folder. Finally, please remove their legends from the Appendix file.
- The movie file needs to be renamed to 'Movie EV1' and the corresponding callouts updated accordingly. Please remove the legend from the and provide it as a README file ZIPped together with the movie file.
- Please add callouts to the reagents and tools table in the methods section.
- The bacterial images shown in Fig. 1A are also shown in Appendix Fig. S6A and S6B. Please clearly mention and explain this re-sue in the respective figure legends.

In addition, I would need from you uploaded separately:

I look forward to seeing the final revised version of your manuscript when it is ready.

Please let me know if you have questions regarding the revision.

Best,

Referee #1:

The authors have appropriately addressed my concerns and suggestions. In my opinion, the manuscript can be accepted in its current form.

All editorial and formatting issues were resolved by the authors.

Dr. Thomas Boesen
Aarhus University
Department of Molecular Biology and Genetics
Gustav Wieds Vej 10
Aarhus C 8000
Denmark

Dear Dr. Boesen,

I am very pleased to accept your manuscript for publication in the next available issue of EMBO reports. Thank you for your contribution to our journal.

Yours sincerely,
